# Endocannabinoids and their receptors modulate endometriosis pathogenesis and immune response

Harshavardhan Lingegowda[1], Katherine B Zutautas[1], Yuhong Wei[2], Priyanka Yolmo[1,3], Danielle J Sisnett[1], Alison McCallion[1], Madhuri Koti[1,3], Chandrakant Tayade[1]*

[1]Department of Biomedical and Molecular Sciences, Queen's University, Kingston, Canada; [2]Rosalind and Morris Goodman Cancer Institute, McGill University, Montreal, Canada; [3]Division of Cancer Biology and Genetics, Queen's University, Kingston, Canada

**Abstract** Endometriosis (EM), characterized by the presence of endometrial-like tissue outside the uterus, is the leading cause of chronic pelvic pain and infertility in females of reproductive age. Despite its high prevalence, the molecular mechanisms underlying EM pathogenesis remain poorly understood. The endocannabinoid system (ECS) is known to influence several cardinal features of this complex disease including pain, vascularization, and overall lesion survival, but the exact mechanisms are not known. Utilizing CNR1 knockout (k/o), CNR2 k/o, and wild-type (WT) mouse models of EM, we reveal contributions of ECS and these receptors in disease initiation, progression, and immune modulation. Particularly, we identified EM-specific T cell dysfunction in the CNR2 k/o mouse model of EM. We also demonstrate the impact of decidualization-induced changes on ECS components, and the unique disease-associated transcriptional landscape of ECS components in EM. Imaging mass cytometry (IMC) analysis revealed distinct features of the microenvironment between CNR1, CNR2, and WT genotypes in the presence or absence of decidualization. This study, for the first time, provides an in-depth analysis of the involvement of the ECS in EM pathogenesis and lays the foundation for the development of novel therapeutic interventions to alleviate the burden of this debilitating condition.

*For correspondence:
tayadec@queensu.ca

Competing interest: The authors declare that no competing interests exist.

## eLife assessment

This study presents **valuable** findings on how the endocannabinoid system is involved in endometriosis progression using CNR1 and CNR2 knockout (KO) mouse models. The evidence supporting the authors' claims is **incomplete**; including bulk RNA-seq, flow cytometry, and imaging mass cytometry would have strengthened the study. This work might be of interest to medical scientists working on endometriosis.

## Introduction

EM is a chronic gynecological disorder characterized by the presence and growth of endometrial-like tissue outside the uterus, referred to as ectopic lesions. Despite its global impact on approximately 200 million individuals and the profound reduction in their quality of life, the exact origins of EM remain elusive (*Giudice and Kao, 2004*). Accumulating evidence, including our previous studies, highlights that components of the ECS are dysregulated within the EM lesion microenvironment as well as in the systemic circulation of EM patients (*Lingegowda et al., 2021b*; *Bilgic et al., 2017*; *Shen et al.,*

*2019*). The ECS is a complex signaling network comprised of canonical receptors (CNR1 and CNR2) and endocannabinoid (EC) ligands, along with a non-canonical extended signaling network of ligands and enzymes extensively reviewed elsewhere (*Di Marzo et al., 2004*). CNR1 and CNR2 are primarily expressed in nerve tissues, immune cells, and reproductive tissues, where they regulate various physiological processes, including pain perception, immune responses, and reproductive functions (*Zou and Kumar, 2018*). Consequently, EM pathogenesis has been postulated as a consequence of EC deficiency (*Russo, 2016*; *Lingegowda et al., 2022*).

Even though the precise etiology of EM is not known, the widely accepted Sampson's theory of retrograde menstruation suggests that EM lesions originate from refluxed, endometrial fragments deposited during menstruation *Sampson, 1927*. Both pregnancy and menstruation depend on spontaneous decidualization of endometrial stroma that is extensively remodeled under the influence of hormones, growth factors, and select cytokines that orchestrate immune cell recruitment and vascular adaptions (*Gellersen and Brosens, 2014*). There is clear evidence that EM patients have defects in eutopic endometrium, including differential expression of key endometrial receptivity markers such as leukemia inhibitory factor (*LIF*), protein arginine methyltransferase 5 (*PRMT5*), and homeobox protein hox-A10 (*HOXA10*), that have been associated with EM and subsequent infertility (*Cai et al., 2022*; *Tomassetti and D'Hooghe, 2018*; *Zutautas et al., 2023*). Evidence also suggests that components of the ECS, including CNR1 and CNR2, are important in maintaining tissue integrity during decidualization and successful implantation of the embryo. Indeed, several reports indicate that mice lacking cannabinoid receptors, CNR1 and CNR2, displayed impaired implantation, increased pregnancy failure rates, heightened edema, and inadequate primary decidual zone formation, highlighting the crucial role of ECS signaling in successful decidualization, implantation, and pregnancy (*Wang et al., 2008*; *Li et al., 2019*; *Sun et al., 2010*).

In EM, CNR1 and CNR2 activation aids in controlling lesion proliferation, pain, and vascularization (*Lingegowda et al., 2021a*; *Leconte et al., 2010*). Keeping in view dysregulated ECS signaling and their central role in decidualization and fertility, we hypothesize that altered CNR1 and CNR2 expression will disrupt ECS signaling dynamics, leading to further lesion development. Furthermore, the involvement of ECS in modulating immune response and homeostasis, may disrupt the immune dynamics and foster lesion establishment.

We conducted a comprehensive investigation into the role of the dysregulated ECS in EM establishment and progression by utilizing CNR1 k/o and CNR2 k/o mouse models. To address the underlying causes of ECS dysfunction, we induced artificial decidualization in WT, CNR1 k/o, and CNR2 k/o mice and used the endometrial fragments from decidualized (DD) and undecidualized (UnD) uterine horns to induce EM in recipient mice of their respective genotypes. Furthermore, we explored the immunomodulatory potential of the ECS in EM, shedding light on how alterations in EC signaling may influence immune cell behavior within the localized peritoneal milieu in mice induced with EM. Our study contributes to the foundational knowledge around ECS dysregulation in EM and paves the way for potential therapeutic strategies targeting ECS for disease management.

## Results

### Ligands of the ECS are dysregulated in a mouse model of EM lacking CNR1 and CNR2 receptors

Based on our previous work demonstrating dysregulated ligands of the ECS in both patients and our mouse model of EM (*Lingegowda et al., 2021b*), we first evaluated whether the absence of CNR1 or CNR2 led to ECS ligand alterations. To do this, we performed targeted mass spectrometry on plasma and EM lesions obtained from CNR1 k/o, CNR2 k/o, and their WT controls. In these mice, EM was induced using UnD and DD tissues obtained from their respective strains into matched recipients (for example, UnD and DD from CNR1 k/o donor mice was implanted into CNR1 k/o recipient mice). We detected some of the major EC ligands such as, 2-Arachidonoylglycerol (2-AG), N-arachidonoylethanolamine (AEA), Palmitoylethanolamide (PEA), and Oleoylethanolamide (OEA) in plasma and EM lesions from all genotypes. All identified ECS ligands are predominantly anti-inflammatory and the range of 2-AG, AEA, PEA, and OEA in the plasma and lesions were comparable to our previous study (*Lingegowda et al., 2021b*). In the plasma, we found no significant differences in ECS ligands across all groups (*Figure 1C–F*), which could be due to the rapid homeostasis achieved in circulation

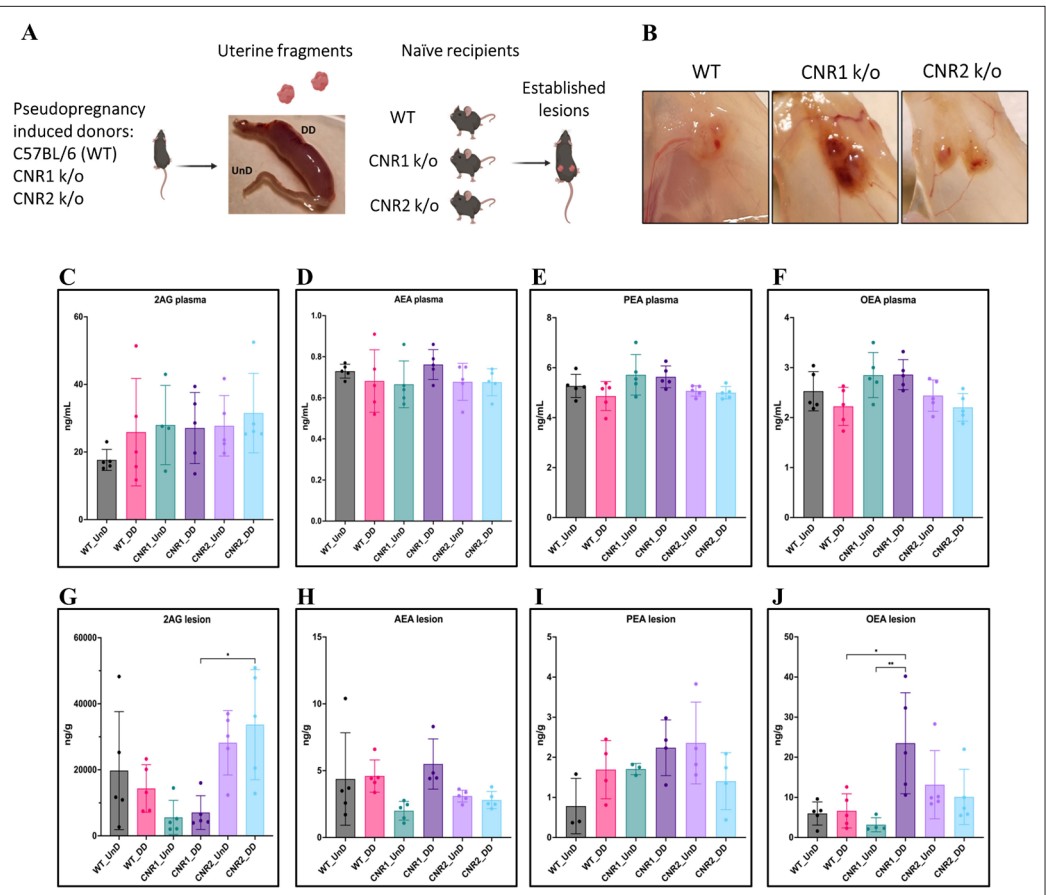

**Figure 1.** Characterization of endocannabinoid ligands in a modified syngeneic mouse model of endometriosis. (**A**) Overview of the modified syngeneic mouse model of endometriosis (EM) where pseudo-pregnant wild-type (WT), CNR1 k/o, and CNR2 k/o mice were induced with decidualized (DD) by injecting sesame oil into the lumen of one uterine horn and the contralateral horn served as undecidualized (UnD) control. Two, 3 mm UnD and DD harvested fragments were implanted into their respective recipient mouse strain to induce EM. (**B**) Representative images of the EM lesions from WT, CNR1 k/o, and CNR2 k/o mice retrieved from the peritoneal cavity at end point (7 days post EM induction surgery). (**C–J**) Bar plots (mean ± SD) showing the concentration of endocannabinoid (EC) ligands 2-Arachidonoylglycerol (2AG), arachidonoylethanolamine (AEA), Palmitoylethanolamide (PEA), and oleoylethanolamide (OEA) identified in the plasma and EM lesions from mice using targeted liquid chromatography-mass spectrometry (LC-MS) approach. (**C–F**) 2AG, AEA, PEA, and OEA were detected in plasma samples without any significant differences between groups. (**G, J**) Significantly higher concentration of 2AG was observed between the DD lesions of CNR2 k/o and CNR1 k/o mice, and significantly higher levels of OEA in the DD lesions from CNR1 k/o mice compared to DD lesions from WT mice. (**H, I**) AEA and PEA levels in the tissue samples did not differ significantly between the comparison groups. n=4–5 individual biological samples per genotype. Statistical analyses were performed using the ordinary one-way ANOVA with Holm-Sidak post hoc test. *p<0.05 and **p<0.01.

(**Lu and Mackie, 2016**). However, in the lesion microenvironment, we captured higher levels of several EC ligands (**Figure 1G–J**). In CNR1 k/o mice, significantly higher concentrations of OEA were observed in the DD compared to UnD lesions (**Figure 1J**), and overall, was on average twofold higher compared to both lesions from WT and CNR2 k/o mice. 2-AG, which selectively binds to the CNR2 receptor was significantly higher in both the UnD and DD EM lesions from CNR2 k/o mice (**Figure 1G**) compared to the CNR1 k/o counterparts. This could indicate a compensatory response in the absence of CNR2. Together, these findings provide insights into the potential dysregulation of ECS ligands in the absence of CNR1 and CNR2 and their involvement in the DD vs UnD scenario during EM lesions establishment.

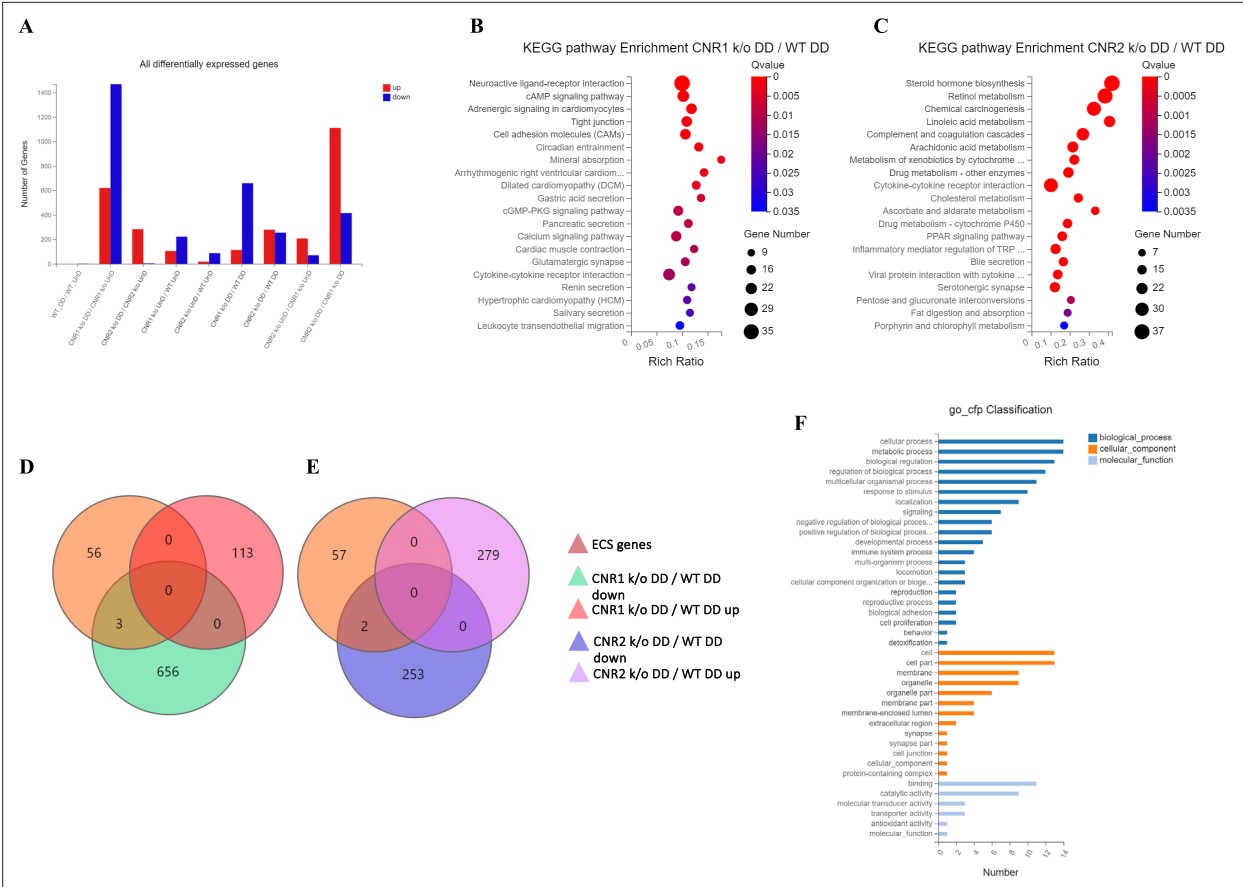

**Figure 2.** Transcriptomic profiling of endometriosis-like lesions from canonical receptors (CNR1 and CNR2) knockout mice reveals extensive differential gene expression and altered pathways. (**A**) Summary of the differentially expressed genes (DEGs) from bulk RNA sequencing analysis conducted on both undecidualized (UnD) and decidualized (DD) lesions from wild-type (WT), CNR1 k/o, and CNR2 k/o mice, revealing extensive changes in gene expression profiles among the different genotypes and lesion types. A total of 1100 and 639 DEGs were identified in both UnD and DD lesions of CNR1 k/o and CNR2 k/o mice, respectively, compared to WT controls. (**B**) Kyoto Encyclopedia of Genes and Genomes (KEGG) pathway analysis revealed significantly altered cell adhesion molecules and cyclic adenosine monophosphate (cAMP) signaling pathways in DD lesions of CNR1 k/o. (**C**) KEGG pathway analysis in DD lesions of CNR2 k/o mice showed changes associated with cytokine receptor interactions and steroid hormone biosynthesis pathways. (**D, E**) Venn diagrams showing the DEGs among the 59 genes directly associated with the endocannabinoid system (ECS), where we found limited DEGs in DD lesions of CNR1 k/o (3) and CNR2 k/o mice (2), respectively. (**F**) A comprehensive gene ontology analysis highlighting the roles of 59 ECS genes across diverse biological processes (blue), cellular (orange), and molecular functions (light blue), accentuating their broader impact beyond canonical ECS functions. Gene Number indicates the number of DEGs enriched in pathway. Rich Ratio indicates the ratio of enriched DEGs to background genes and Q-value indicates significance, with a value closer to zero being more significant and is corrected by Benjamini-Hochberg method.

The online version of this article includes the following figure supplement(s) for figure 2:

**Figure supplement 1.** Differentially expressed genes (DEGs) of bulk RNA sequencing between the undecidualized (UnD) lesions from CNR1 k/o and CNR2 k/o compared to wild-type (WT) controls.

## Impact on gene expression and pathway alterations in EM lesions from mice in the absence of CNR1 and CNR2

Next, we investigated the effects of CNR1 and CNR2 absence on the transcriptomic profile of both UnD and DD EM lesions from their respective genotypes. Bulk RNA sequencing was performed on both UnD and DD lesions from WT, CNR1 k/o, and CNR2 k/o mice as detailed earlier to elucidate the molecular alterations associated with the disruption of these two-receptors signaling. Differential expression analysis revealed changes in gene expression profiles among the different genotypes and lesion types (*Figure 2A*). A total of 1100 and 639 differentially expressed genes (DEGs) were found in both UnD and DD lesions of CNR1 k/o and CNR2 k/o mice, respectively, compared to WT controls (UnD data is provided in *Supplementary file 2*). To gain insights into the biological implications of

the observed gene expression changes, we conducted a Kyoto Encyclopedia of Genes and Genomes (KEGG) pathway enrichment analysis on the DEGs identified in UnD and DD lesions of CNR1 k/o and CNR2 k/o mice compared to WT mice. In the DD lesions from CNR1 k/o mice, KEGG pathway analysis revealed significant alterations in several pathways (*Figure 2B*). Notably, the cell adhesion molecules pathway was prominently affected, indicating a potential role for CNR1 in mediating cell-cell interactions and tissue remodeling processes. Additionally, the cyclic adenosine monophosphate (cAMP) signaling emerged as another negatively impacted pathway, implicating CNR1 in modulating intracellular signaling cascades. In DD lesions from the CNR2 k/o mice, the analysis highlighted distinct pathways affected in the context of inflammation and EM (*Figure 2C*) including the cytokine receptor interactions pathway, pointing to the involvement of CNR2 in immune responses and inflammatory processes associated with EM. Furthermore, we captured alterations in the steroid hormone biosynthesis pathway suggesting a role for CNR2 in hormone-related mechanisms relevant to endometrial tissue development and homeostasis.

Next, we performed a subset analysis for genes directly involved in ECS signaling. A total of 59 key genes in ECS signaling were selected, as identified in a study by *Tanaka et al., 2022*. Surprisingly, despite the central role of CNR1 and CNR2 in ECS signaling, we found a limited number of DEGs related to this system. Out of 59 genes directly associated with ECS, only 3 (*Cnr1, Plch1,* and *Plch2*) and 2 (*Cnr2* and *Plag2ge*) DEGs were identified in the DD lesions of CNR1 k/o (*Figure 2D*) and CNR2 k/o (*Figure 2E*), respectively, compared to DD lesions from WT controls. This result suggests that CNR1 and CNR2 modulate the EM microenvironment through intricate interactions with other signaling pathways beyond the canonical ECS pathway. A comprehensive gene ontology (GO) classification analysis on the 59 identified ECS genes (*Figure 2F*) unveiled their multifaceted roles in reproductive functions, immune system regulation, and cellular processes. The genes exhibited enrichment in molecular functions such as receptor activity and lipid binding. In terms of cellular components, these genes were associated with plasma membrane structures, and intracellular compartments, signifying their diverse subcellular localization and potential involvement in dynamic cellular processes. Furthermore, GO analysis highlighted their participation in biological processes such as immune response modulation, lipid metabolism, cell communication, and intracellular signaling pathways, indicating the broader impact of ECS beyond its canonical functions. The results of the UnD comparisons between the genotypes are provided in the supplementary data (*Figure 2—figure supplement 1*). While the UnD lesions exhibited distinct gene expression patterns compared to DD lesions, common trends in the effects of CNR1 k/o and CNR2 k/o on gene expression were observed across both lesion types. Specifically, the cytokine receptor interaction, complement cascade, and inflammatory mediator pathways were significantly altered in the UnD lesions from CNR1 k/o and CNR2 k/o mice compared to UnD lesions from WT mice. Overall, our findings highlight the significant impact of CNR1 and CNR2 k/o on gene expression in EM lesions and the implications for EM pathogenesis.

## Disruption of genes related to the adaptive immune response in EM lesions without CNR1 and CNR2

Building upon our previous investigation into the transcriptomic alterations, we conducted an in-depth analysis of differentially expressed immune-related genes (as per InnateDB version 5.4) in both UnD and DD lesions across all genotypes. Here, our analysis is focused on the immune-related genes within DD lesions of CNR1 k/o and CNR2 k/o mice compared to WT controls. Comparison of the UnD lesions of CNR1 k/o and CNR2 k/o with WT EM mice are included in the supplementary data (*Figure 3—figure supplement 1*). The differential expression bar plot (*Figure 3A*) provides a representation of the upregulated and downregulated genes in each comparison. The volcano plots for DD lesions from CNR1 k/o vs. WT (*Figure 3B*) and CNR2 k/o vs. WT (*Figure 3C*) illustrate the various DEGs. Notably, CNR1 k/o DD lesions exhibited 39 downregulated (e.g. *Nlrp6* and *Il1a*, pivotal regulators of inflammatory response) and 14 upregulated genes (e.g. *Cxcl9* and *Cxcl10*, chemokines involved in immune cell recruitment), while CNR2 k/o DD lesions showed 40 downregulated (e.g. *Siglecg* and *Il6*, involved in immune regulation and pro-inflammatory response) and 25 upregulated genes (e.g. *C8a*, *C9*, and *Masp2*, part of the complement system), highlighting substantial changes in gene expression associated with CNR1 and CNR2 disruption compared to the DD lesions from WT mice. Of particular interest, we observed significant downregulation of T cell-related genes (*Cd3e*, *Cd3g*, *Gata3*, and *Ctla4*) in the CNR2 k/o DD lesions (*Supplementary file 3*), aligning with the CD3 + T cell dysfunction

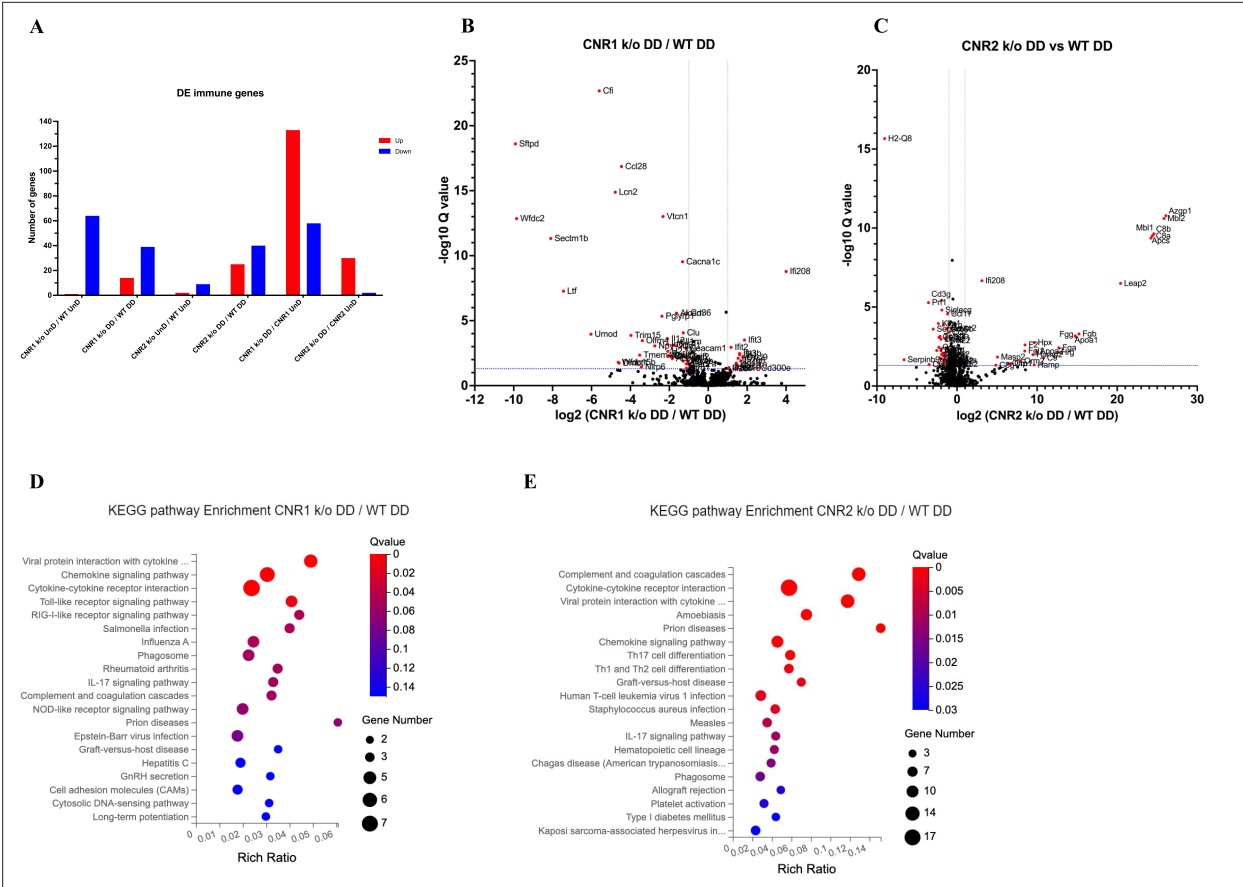

**Figure 3.** Bulk RNA sequencing revealed alterations in immune-related gene expression and pathway in endometriosis (EM) lesions from CNR1 k/o and CNR2 k/o mice. (**A**) Bar plot overview of the differentially expressed (DE) immune-related genes among different genotypes and lesion types. (**B, C**) The volcano plots for decidualized (DD) lesions of CNR1 k/o vs. wild-type (WT) and CNR2 k/o vs. WT, respectively, revealed 39 downregulated and 14 upregulated genes in CNR1 k/o DD lesions, while CNR2 k/o DD lesions exhibited 40 downregulated and 25 upregulated genes. Log2 fold change as the x-axis and log10 Q-value (FDR adjusted) as the y-axis. Vertical dotted lines on the x-axis indicate ±onefold change and vertical dotted line on the y-axis indicate Q-value of 0.05. (**D, E**) Kyoto Encyclopedia of Genes and Genomes (KEGG) pathway analysis of differentially expressed genes (DEGs) in CNR1 k/o DD lesions show significant alteration in the chemokine signaling pathway, cytokine-cytokine receptor interaction, and toll-like receptor signaling pathways, while in CNR2 k/o DD lesions, alterations were observed in pathways related to cytokine-cytokine receptor interaction, Th17, Th1, and Th2 cell differentiation.

The online version of this article includes the following figure supplement(s) for figure 3:

**Figure supplement 1.** Kyoto Encyclopedia of Genes and Genomes (KEGG) pathway analysis of immune specific differentially expressed genes (DEGs).

observed in the PF and spleen and further validations from in-vitro functional assay (mentioned below). However, we did not find the same differences (T cell-related genes) in the UnD lesions of CNR2 k/o mice. Moreover, UnD lesions of CNR2 k/o mice showed a significantly low number of DEGs (11 compared to 65 in the DD lesions from CNR2 k/o mice) suggesting a decidualization-dependent response (*Supplementary file 3*). This observation clearly emphasizes a potential link between CNR2 dysfunction with decidualization characterized by T cell signaling issues within the EM microenvironment. To understand the functional implications of the DEGs, we conducted KEGG pathway analysis on specific differentially expressed immune genes in DD lesions from CNR1 k/o and CNR2 k/o mice. Notably, in CNR1 k/o DD lesions compared to WT DD, the chemokine signaling pathway, cytokine-cytokine receptor interaction, and toll-like receptor (TLR) signaling pathways were negatively affected (*Figure 3D*). These findings provide insights into the impact of CNR1 disruption on immune cell function within the DD environment. Conversely, CNR2 k/o DD lesions were associated with significant alterations in cytokine-cytokine receptor interaction, Th17, Th1, and Th2 cell differentiation pathways (*Figure 3E*). These pathways are proven to be involved in T cell development, differentiation, and effector functions, aligning with our observed dysregulation of T cell-related genes. Together, these

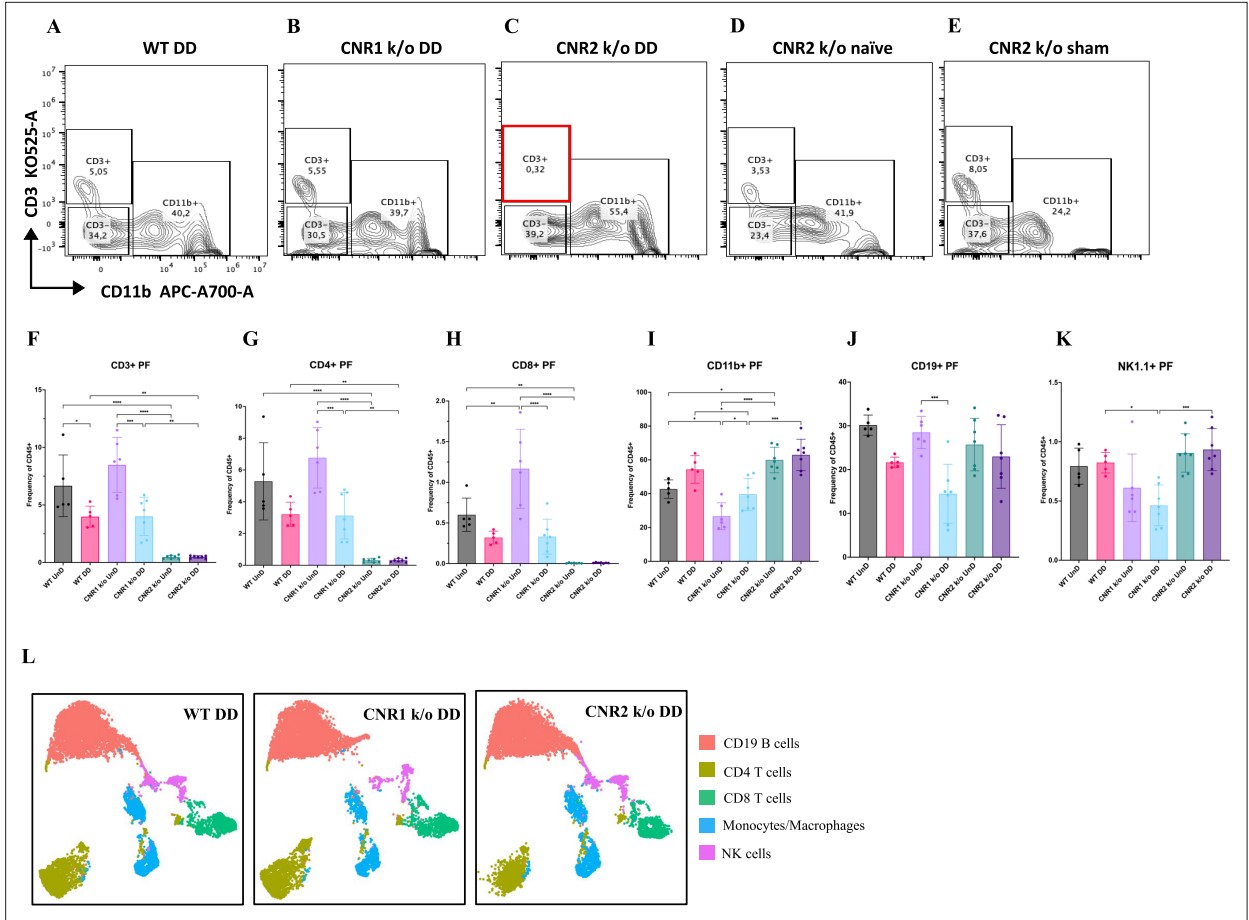

**Figure 4.** Flow cytometry profiling of peritoneal fluid (PF) and splenic cells show altered immune cell phenotypes in CNR1 k/o and CNR2 k/o mice with endometriosis (EM). (**A–E**) Gating panels of PF cells showing CD3 KO525-A on the y-axis vs CD11b APC-A700-A on the x-axis among wild-type (WT), CNR1 k/o, and CNR2 k/o mice with decidualized (DD) EM lesions, as well as CNR2 k/o naive and CNR2 k/o sham-operated controls. (**F**) CD3 + total T cells in the PF of CNR2 k/o mice with EM lesions, regardless of lesion types, were significantly reduced compared to WT and CNR1 k/o mice with EM, as well as CNR2 k/o naïve and sham operated mice. (**G, H**) This extended to the subsets of CD3 + T cells, CD4+, and CD8 + T cells, respectively. CNR1 k/o mice with DD lesions also exhibited significantly decreased CD3 + total T cells, CD4 + helper T cells, and CD8 + cytotoxic T cell frequencies compared to their undecidualized (UnD) counterparts. (**I**) CD11b+ monocyte/macrophage populations were increased in the PF of CNR2 k/o mice with UnD and DD lesions compared to WT and CNR1 k/o mice. CNR1 k/o mice with DD lesions displayed higher monocyte/macrophage populations compared to their UnD counterparts. (**K, J**) CNR1 k/o mice with DD lesions exhibited lower CD19 + B cells and NK1.1+ NK cell populations compared to WT and CNR2 k/o mice. (**L**) Immune cell populations in splenocytes were analogous to findings from PF cells, depicted by tSNE plots. n=5–7 individual biological samples per genotype. Statistical analyses were performed using the ordinary one-way ANOVA with Holm-Sidak post hoc test. *p<0.05, **p<0.01, ***p<0.001, and ****p<0.0001. Data presented as mean ± SD.

The online version of this article includes the following figure supplement(s) for figure 4:

**Figure supplement 1.** Gating panel representing T cells from splenocytes.

**Figure supplement 2.** Flow cytometry analysis of immune cell phenotypes in the splenocytes of wild-type (WT), CNR1 k/o and CNR2 k/o mice with EMS.

findings further elucidate the roles of CNR1 and CNR2 in modulating immune responses within the context of EM.

## Multispectral flow cytometry revealed altered immune cell profiles in a mouse model of EM lacking CNR1 and CNR2

Immune dysregulation is recognized as a crucial factor in the pathogenesis of EM (*Symons et al., 2018*). To elucidate the impact of CNR1 and CNR2 absence on immune cell populations, we performed multispectral immune profiling in splenocytes and cells from the PF from WT, CNR1 k/o, and CNR2 k/o mice harboring UnD and DD lesions. Representative gating panels of PF CD3 (y-axis) vs CD11b

(x-axis) cells (*Figure 4A–E*) illustrate distinct profiles among WT, CNR1 k/o, and CNR2 k/o mice with EM, as well as CNR2 k/o naïve, non-operated mice, and CNR2 k/o sham-operated controls. Strikingly, CD3 + T cell populations were nearly absent in the PF of CNR2 k/o mice with EM, regardless of lesion types (UnD and DD), when compared to other groups (*Figure 4C and F*). This trend further extended to CD4 + helper T cells and CD8 + cytotoxic T cells (*Figure 4G–H*). CNR1 k/o mice with DD lesions exhibited significantly reduced CD3+ (*Figure 4F*), CD4+ (*Figure 4G*), and CD8+ (*Figure 4H*) T cell frequencies compared to their UnD counterparts, as well as lower CD19 + B cells and NK1.1+ NK cells (*Figure 4K*) populations, compared to WT and CNR2 k/o mice. Concomitant with the reduction of T cell subsets, an increase in CD11b monocytes was observed in the PF of CNR2 k/o mice with UnD and DD lesions compared to WT and CNR1 k/o mice (*Figure 4I*). Similarly, CNR1 k/o mice with DD lesions displayed higher monocyte/macrophage populations compared to their UnD counterparts (*Figure 4I*). Furthermore, WT mice with DD lesions demonstrated significantly lower CD3 + T cell frequencies compared to their UnD counterparts (*Figure 4F*), suggestive of a decidualization-associated effect. Splenocytes exhibited analogous trends (*Figure 4L*), as depicted by tSNE plots (bar plots and contour gating plots in *Figure 4—figure supplements 1 and 2*). These alterations in immune cell numbers reinforce the influence of CNR1 and CNR2 dysregulation and decidualization on immune cell populations, confirmed both locally in PF and systemically in the spleen.

## T cells from CNR2 k/o mice exhibit impaired viability upon TCR activation

To validate the functional consequences of CNR2 deficiency on T cell behavior, we conducted a series of in-vitro assays using T cells isolated from splenocytes of naïve WT and CNR2 k/o mice. CD3 + T cells were activated non-specifically, with or without PMA/ionomycin cocktail, in the presence or absence of tumor necrosis factor-alpha (TNFα) to create a sterile inflammatory challenge.

We observed a significant reduction in the viability of total CD3 + T cells from CNR2 k/o mice upon activation with PMA/ionomycin (gating strategies in *Figure 5—figure supplement 1*) compared to media controls (*Figure 5A and B*). In contrast, WT CD3 + T cells activated with PMA/ionomycin, with or without TNFα, exhibited no significant difference in viability when compared to the media control (*Figure 5A and B*). This observation aligns with our in-vivo findings, whereby CD3 + T cells from CNR2 k/o mice with EM exhibited a significant reduction in both the splenic and PF population, but not in the SHAM-operated mice, emphasizing the EM-specific nature of this effect. Additionally, the overall reduced viability of CD3 + T cells of CNR2 k/o mice upon PMA/ionomycin activation led to a decrease in the proliferation of CD4 + T cells (*Figure 5C*) but not of CD8 + T cells (*Figure 5D*). However, our results indicated that although CNR2-deficient T cells displayed reduced viability upon activation, they exhibited higher levels of IFNγ production compared to CD3 + T cells from WT mice, suggesting their functional competence (*Figure 5—figure supplement 1*). These findings shed light on the intricate role of CNR2 in modulating T cell responses, with potential implications for immune dysregulation.

## Spatial cell analysis of EM lesions from WT, CNR1 k/o, and CNR2 k/o mice reveal altered immune cell populations

To gain a comprehensive insight into the spatial distribution of immune cells and stromal cell types within EM lesion architecture of UnD and DD lesions of WT, CNR1 k/o, and CNR2 k/o mice, we employed IMC analysis. This approach aimed to elucidate the impact of CNR1 and CNR2 absence on the cellular composition and organization of EM lesions in mice. The schematic representation of the IMC procedure (*Figure 6A*) outlines the steps involved in this analysis. Following the acquisition of two regions of interest (ROI) per section (based on the H&E stain), single-cell segmentation (*Figure 6B*) and subsequent segmentation quality assessment (*Figure 6C*) were performed. After batch effect correction of samples, non-linear dimensionality reduction of the sample type revealed a distinct expression pattern of immune cells and cell state markers between UnD and DD lesions, as well as differences between the genotypes (*Figure 6D*). After unsupervised phenotyping and labeling of the different cell types, uniform manifold approximation and projection (UMAP) dimensionality reduction further highlighted the key differences in cell composition between UnD and DD lesions (*Figure 6E*). Overall, combined expression of the cell types of DD lesions from all three genotypes exhibited increased stromal compartments, decreased epithelial cells, and heightened macrophage

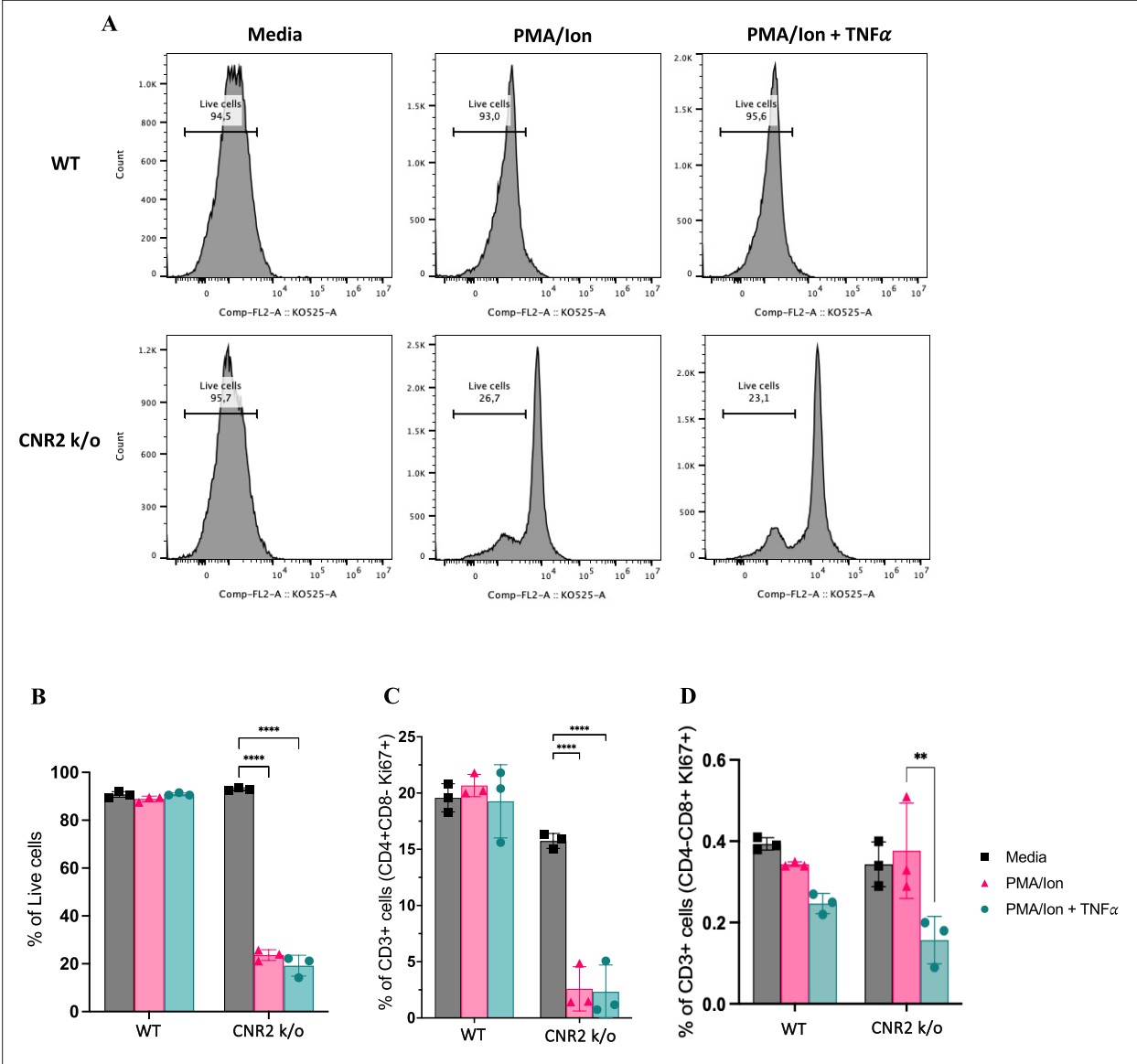

**Figure 5.** In-vitro validation of CNR2 deficiency on CD3 T cell viability and functionality in conditions representative of Endometriosis (EM) lesion microenvironment. (**A**) Representative gating of percentage live, CD3 + total T cells from wild-type (WT) and CNR2 k/o mice activated with or without PMA/Ionomycin cocktail in the presence or absence of tumor necrosis factor-alpha (TNFα) and media control. Graph with live gating shows count on the y axis and live/dead-KO525 marker on the x-axis. (**B**) Bar graphs of percentage live population of CD3 + total T cells from CNR2 k/o mice show a significant decrease in the viability of cells activated with PMA/ionomycin with or without TNFα. Whereas, no significant changes were observed in the CD3 + total T cells from WT mice as well as from CNR2 k/o mice in media. (**C**) Activation of CD3 + T cells of CNR2 k/o mice with PMA/ionomycin affected proliferation of CD4 + helper T cells specifically, with or without the presence of TNFα when compared to both their media control and WT controls. (**D**) No significant changes were observed in the proliferative CD8 + cytotoxic T cells from CNR2 k/o mice compared to their WT controls across different activation and non-activation groups. Ordinary two-way ANOVA with Tukey's post hoc test was performed to assess statistical significance. **$p < 0.05$, ****$p < 0.0001$. n=3 technical replicates. Data presented as mean ± SD.

The online version of this article includes the following figure supplement(s) for figure 5:

**Figure supplement 1.** In vitro functional assay and flow cytometry evaluation of activated CD3 T cells from naïve wild-type (WT) and CNR2 k/o mice.

infiltration compared to the expression of cell types from UnD lesions. Representative images illustrate the distribution of different cell types based on unsupervised clustering and labeling (*Figure 7A*).

Further analysis of the cell type distribution (*Figure 7B*) through bar plots unveiled several differences. Although not significant, due to the relatively low number of biological replicates, T cell expression was increased (CD4 + helper T cells and CD8 + cytotoxic T cells) in both UnD and DD lesions of

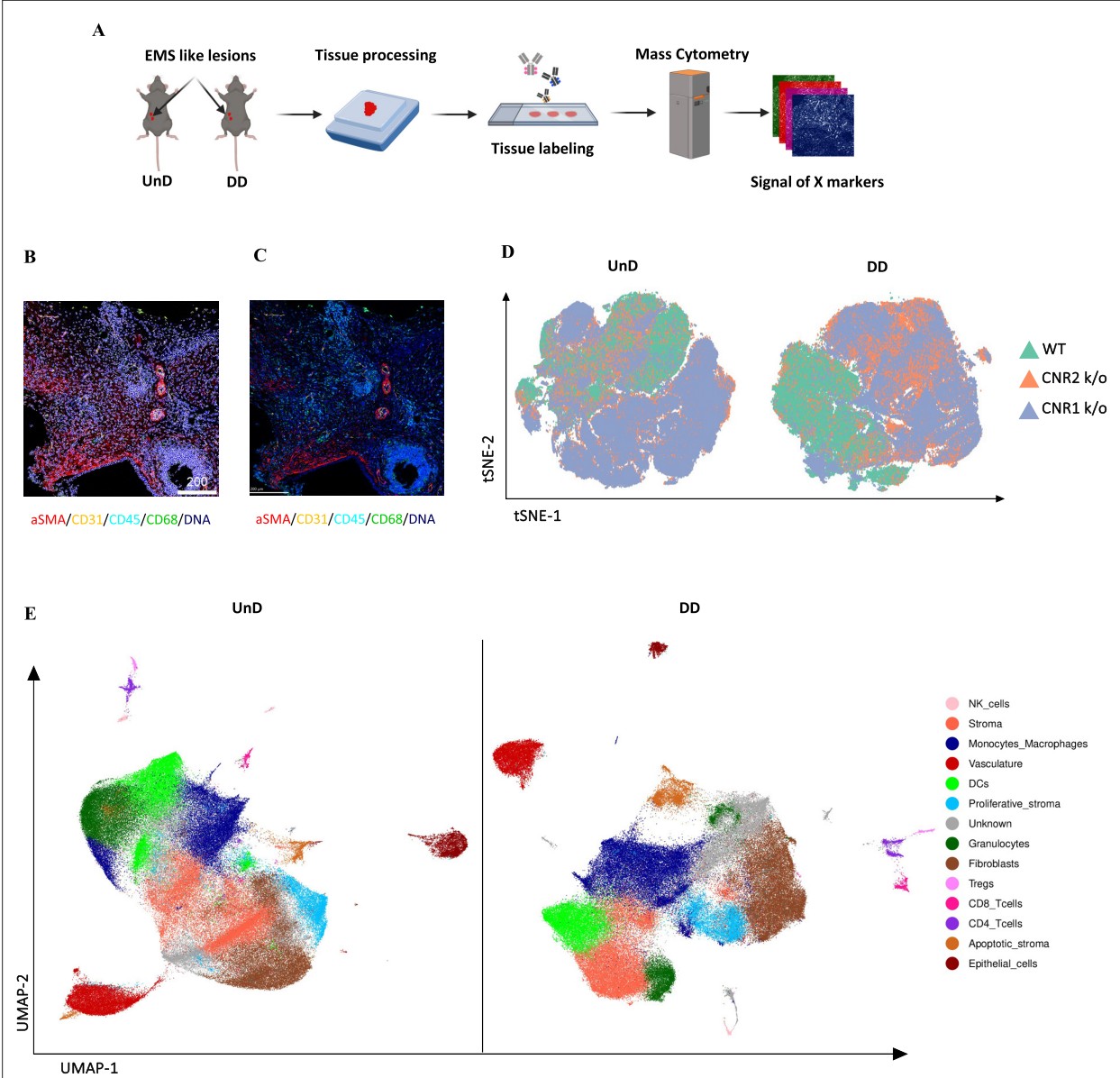

**Figure 6.** Imaging mass cytometry spatial profiling of immune cell distribution and cellular patterns in endometriosis (EM) lesions in CNR1 k/o, CNR2 k/o, and wild-type (WT) mice. (**A**) The imaging mass cytometry (IMC) data collection and analysis workflow outlines the steps involved in gaining comprehensive insights into the spatial distribution of immune cells and relevant cell types within undecidualized (UnD) and decidualized (DD) EM-like lesions of WT, CNR1 k/o, and CNR2 k/o mice. (**B, C**) Representative images showing the single-cell segmentation performed following the acquisition of two regions of interest (ROI) per section (three biological samples per genotype) and segmentation quality of the data after segmentation analysis was conducted, respectively. (**D**) Non-linear dimensionality reduction after batch effect correction showed distinct expression patterns of immune cells and cell state markers between UnD and DD lesions. DD lesions from the CNR1 k/o and CNR2 k/o mice showed expression pattern that was significantly different from the DD lesions of WT mice, as well as compared to UnD lesions among different genotypes. (**E**) Uniform manifold approximation and projection (UMAP) dimensionality reduction highlighted key cell types and differences in composition between UnD and DD lesions. DD lesions exhibited increased stroma and fibroblasts, decreased epithelial cells, and heightened macrophage infiltration compared to UnD lesions.

CNR1 k/o and CNR2 k/o mice compared to WT (*Figure 7C*). This observation highlights that resident T cells are not impacted in the absence of CNR1 and CNR2 within the endometriotic milieu. Intriguingly, in EM lesions from CNR1 k/o and CNR2 k/o, we saw significantly elevated expression of monocytes/macrophages (*Figure 7D*), stromal cells (*Figure 7E*), and hallmarks of EM such as proliferation and vascularization (*Figure 7F*) demonstrating an altered microenvironment in the absence of these receptors. To comprehend cell-cell interactions and their implications, we conducted a cellular neighborhood

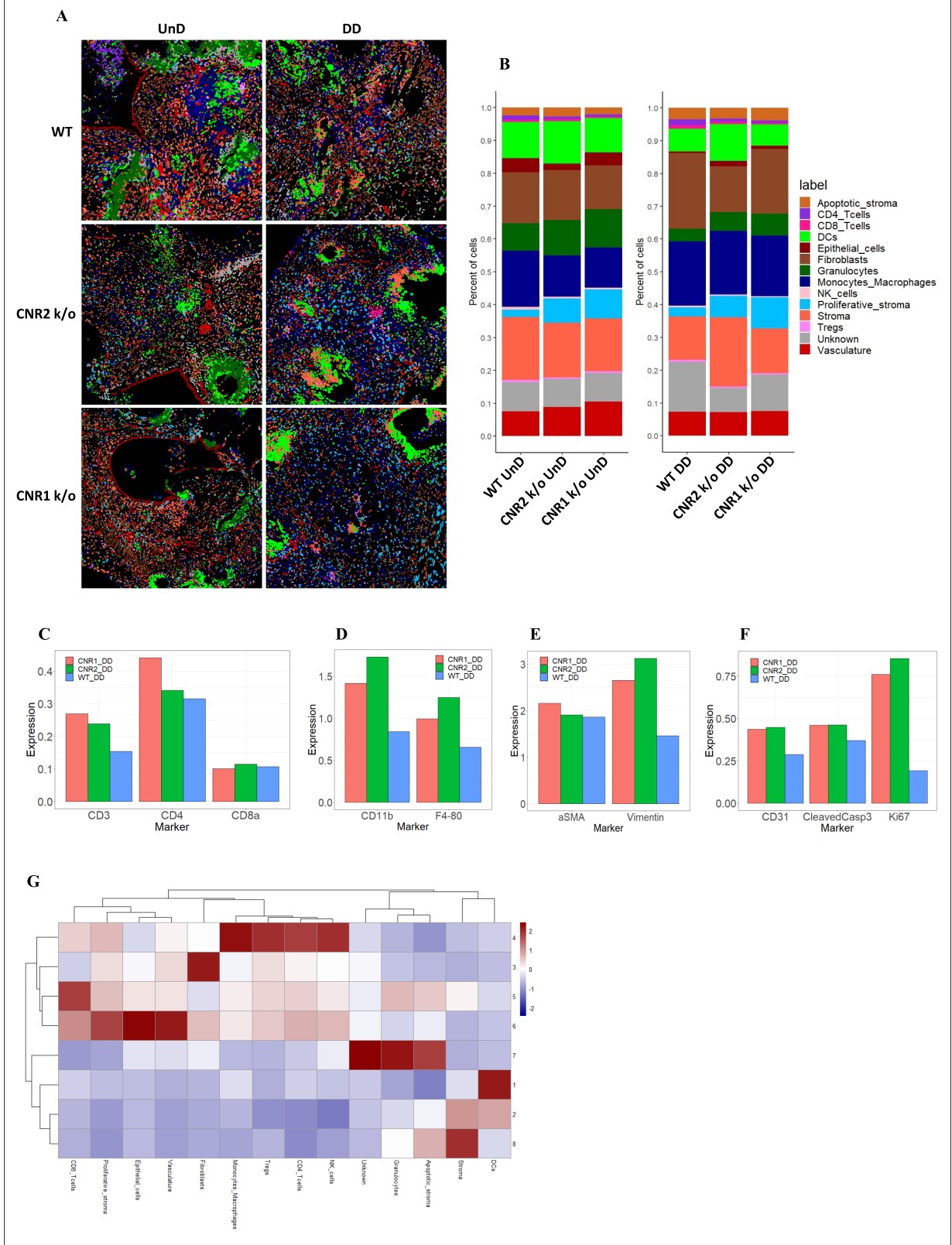

**Figure 7.** Imaging mass cytometry revealed altered cellular composition and neighborhoods in endometriosis (EM) lesions from CNR1 k/o, CNR2 k/o, and wild-type (WT) mice. (**A**) Representative image showing the distribution of different cell types within EM lesions (n=2–3 per tissue type) based on unsupervised clustering and labeling. (**B**) Stacked bar plots reveal the distribution of various cell types within undecidualized (UnD) and decidualized (DD) lesions of CNR1 k/o and CNR2 k/o mice compared to WT mice. (**C**) Increased CD3 + T cells and CD4 + helper T cells expression was found in the

*Figure 7 continued on next page*

*Figure 7 continued*

DD, EM lesions from CNR1 and CNR2 k/o mice compared to WT mice highlighting that T cells residing in the lesions were not affected. (**D**) CD11b+ monocyte and F4/80+ macrophage expression was significantly increased in the DD, EM lesions from mice lacking CNR1 and CNR2 compared to the WT controls. (**E**) Vimentin expressing stromal compartments that predominantly make up the EM lesions were also significantly increased in the DD lesions from CNR1 k/o and CNR2 k/o mice compared to WT mice. (**F**) Hallmark features of EM lesions, such as proliferation (Ki67+) and vascularization (CD31+) were significantly increased in the DD, EM lesions from mice lacking CNR1 and CNR2 compared to the WT controls. Combined, it highlights the effect on early lesion development and further progression through sustained proliferation due to dysregulated CNR1 and CNR2. (**G**) Heatmap representation of the CN analysis show distinct clustering patterns observed in the DD lesions among the different genotypes, where immune cells mainly clustered together in CN 4, while other cell types such as stroma, epithelial cells, and vasculature exhibited distinct clustering patterns across CN 6, 3, and 8, respectively.

The online version of this article includes the following figure supplement(s) for figure 7:

**Figure supplement 1.** Imaging mass cytometry (IMC) and cellular neighbourhoods (CN) in undecidualized (UnD), EMS lesions from CNR1 k/o, CNR2 k/o, and wild-type (WT) mice.

(CN) analysis. This approach grouped cells based on information within their direct spatial vicinity and identified intricate spatial relationships among diverse cell types within the lesion microenvironment. This analysis revealed distinct clustering patterns across different cell types within the lesion architecture of the DD lesions from WT, CNR1 k/o, and CNR2 k/o mice (*Figure 7—figure supplement 1*). Immune cells predominantly clustered together in CN 4, while other cell types (stroma, epithelial cells, and vasculature) exhibited distinct clustering patterns across CN 6, 3, and 8 in DD lesions (*Figure 7G*). Although most of the immune cell types clustered together in the UnD lesions, cell types of the lesion architecture clustered distinctly compared to the DD lesions (*Figure 7—figure supplement 1*). This clustering emphasizes the interplay between immune cells and the broader cellular components of the lesions. In summary, our comprehensive investigation has unveiled intricate spatial relationships among immune cells and diverse cell types within EM lesions in mice. The observed alterations in T cell expression, coupled with stromal dynamics, in CNR1 k/o and CNR2 k/o lesions underscore the pivotal roles of these receptors in shaping the endometriotic microenvironment.

## Discussion

Emerging evidence from our previously reported findings and others, implicated dysregulation of ECS, comprising CNR1 and CNR2 canonical receptors along with their EC ligands, in EM pathophysiology (*Lingegowda et al., 2021b*; *Sanchez et al., 2016*). The ECS is involved in several physiological processes including (but not limited to) pain perception, immune regulation, and reproductive functions *Rodríguez de Fonseca et al., 2005*; CNR1 and CNR2 are expressed in immune cells, nerve tissues, and serve as critical regulators of reproductive processes including decidualization and embryo implantation (*Di Blasio et al., 2012*; *Walker et al., 2019*). The etiology of EM has been speculated to be routed in defective decidualization and retrograde menstruation of the endometrial fragments combined with ECS dysregulation (*Lingegowda et al., 2022*; *Sampson, 1927*; *Maia et al., 2020*). While it is plausible that decreased ECS function influences EM lesion initiation, progression, and severe pain experience, it is not clear whether ECS dysfunction actively contributes to EM pathogenesis, or whether it represents a secondary consequence of alterations occurring within the refluxed endometrial tissue, leading to the establishment of EM lesions.

Keeping this central dogma in view and to provide insights into early events of EM pathogenesis, we induced EM in both CNR1 k/o and CNR2 k/o mice utilizing syngeneic, DD, and UnD uterine endometrial fragments. Absence of CNR1 and CNR2 did not influence systemic levels of ECS ligands but the lesion microenvironment displayed significant changes in the levels of OEA and PEA, suggesting a tissue-specific response.

One intriguing aspect of ECS involvement in EM is its role in decidualization, a process pivotal for uterine receptivity to embryo implantation and successful pregnancy, that may also contribute to EM establishment (*Correa et al., 2016*). Although both CNR1 and CNR2 are active in decidualization, CNR1 may have a more prominent role. Absence of CNR1 and CNR2 shows compromised decidualization in mice in a CNR1-dependent manner validated through in-vitro studies (*Li et al., 2020*). Similarly, in our study, EM lesions (both UnD and DD) from mice lacking CNR1 showed significantly more DEGs (2088), compared to CNR2 (287) and WT (2). Genes essential for decidualization such as

*Igfbp2*, *Bmp3*, *Ptgdr*, *Wnt7a*, and *Esr1* were downregulated in the DD, EM lesions from CNR1 k/o mice compared to their UnD counterparts. This further reinforces the role of CNR1 in the uterine and EM lesion microenvironment, including their role in decidualization response. Moreover, the interplay between CNR1 and CNR2 are important since CNR2 contributes to immunomodulation, which is a key process during decidualization (*Turcotte et al., 2016*; *Taylor et al., 2010*). Even though retrograde menstruation is considered the main mechanism by which endometrial fragments reach the peritoneal cavity and implant to form EM lesions, the retrograded menstrual debris itself does not undergo decidualization. However, some EM lesions in the peritoneal cavity, under the influence of estrogen and progesterone, can undergo decidualization as the lesions can exist in different evolutionary stages, from active red lesions to quiescent white lesions (*Leone Roberti Maggiore et al., 2016*).

Given the complexity of ECS signaling and compensatory mechanisms, we focused our investigation on the immune dysregulation aspect of EM pathophysiology. Our findings suggest that altered ECS dynamics during decidualization disrupt ECS signaling, leading to dysregulation of immune responses and aberrant cellular behavior. Indeed, immune dysfunction is a hallmark of EM (*Symons et al., 2018*; *Ahn et al., 2015*), and ECS could play a crucial role in shaping immune responses, particularly through its impact on T cell function yet there is no clear evidence. Alterations in T cell populations and functions have been associated with EM progression, suggesting their vital role in EM pathogenesis and maintenance (*Tanaka et al., 2017*; *Kyama et al., 2003*). Our flow cytometry analysis revealed significant alterations in immune cell populations in mice bearing EM lesions, with a notable absence of CD3 + T cells, CD4 + helper T cells, and CD8 + cytotoxic T cells specifically in CNR2 k/o mice. Mechanistic in-vitro studies further confirmed an aberrant T cell response in CNR2 k/o mice, as with T cell receptor activation and stimulation there was decreased viability. Combined, our findings show that CNR2 is critical in T cell survival upon TCR activation with pathogen-associated molecular patterns (PAMPs)/danger-associated molecular patterns (DAMPs) or antigen-mediated signals. These findings also shed light on a previously unrecognized role of CNR2 in EM-associated adaptive immune dysfunction given the critical role of T cells in immune surveillance and regulation. Furthermore, speculation of EM being a cause of ECS dysfunction could be of importance since CNR2 was found to be reduced in the lesions of EM patients, as shown by our previous study (*Lingegowda et al., 2021b*). In addition, the bulk RNA sequencing strengthens the finding of dysregulation of T cell-related genes in CNR2 k/o EM lesions. The observed downregulation of T cell-related genes such as *Cd3e*, *Cd3g*, *Gata3*, and *Ctla4* is consistent with the diminished CD3 + T cell populations and highlights the relevance of CNR2 in T cell-mediated immune responses within the endometriotic microenvironment.

KEGG pathway analysis of differentially expressed immune-related genes in CNR2 k/o DD lesions further revealed that Th1, Th2, and Th17 cell differentiation pathways were impacted, and the previous literature confirms the dysregulation of these pathways in EM pathophysiology (*Chang et al., 2023*; *Andreoli et al., 2011*). Additionally, our in-vitro functional assay showing CD4 + helper T cells being affected (proliferation) more than the CD8 + cytotoxic T cell subset further adds to the subset-specific behaviour of CNR2. Our results provide a missing link between the ECS and immune system functioning during EM.

To gain insights into the early features of lesion initiation and establishment, we conducted an IMC analysis of EM lesions across genotypes and different lesion types (DD and UnD). DD lesions from CNR1 and CNR2 k/o mice showed higher T-cell residing in the lesions with increased stromal compartments and monocytes/macrophages population compared to WT lesions. Stromal cells contribute to the early development of EM lesions by promoting inflammation, angiogenesis, fibrosis, and immune modulation (*McKinnon et al., 2022*; *Queckbörner et al., 2020*; *Zondervan et al., 2018*). Additionally, the interaction between macrophages and stromal cells is important in EM, with the NLRP3 inflammasome playing a role in lesion development (*Guo et al., 2021*). Studies have implicated macrophages in EM lesion growth where they support angiogenesis (formation of blood vessels) by producing pro-angiogenic factors such as vascular endothelial growth factor (VEGF) (*Ding et al., 2012*; *Sun et al., 2019*; *McLaren et al., 1996*). Given the combined increase in proliferation, endothelial markers, and monocytes/macrophages in EM lesions from mice potentially indicates that they could modulate the early lesion microenvironment in the event of CNR1 and CNR2 dysregulation. Based on the proportion of these macrophages to certain phenotypes, such as M1 or M2, would dictate lesion development and subsequent progression. Further studies are required to tease out molecular interactions of CNR1 and CNR2 with specific immune cell subsets in a complex EM lesion

microenvironment and determine how it contributes to the establishment of blood supply and lesion survival.

The dysregulation of the ECS, as evidenced by the altered expression of CNR1 and CNR2 in our mouse model of EM, appears to have far-reaching consequences on the cellular and molecular landscape of endometriotic lesions. Since CNR1 is widely expressed in the central nervous system where it regulates pain perception, mood, and other neurological processes, the dysregulation of CNR1 signaling may impact the sensory innervation and pain responses associated with the EM (*Rodríguez de Fonseca et al., 2005*). Additionally, CNR1 has been shown to modulate immune cell function, including the production of pro-inflammatory cytokines *Klein, 2005*. The loss of CNR1 signaling in immune cells infiltrating the endometriotic lesions likely contributes to the altered immune profiles observed in our study. In contrast, CNR2 is predominantly expressed in immune cells, such as macrophages, lymphocytes, and natural killer cells (*Turcotte et al., 2016*). This receptor plays a crucial role in the regulation of immune responses, including the modulation of cytokine production, cell migration, and proliferation. The significant reduction in CD3 + T cells in the peritoneal cavity and spleen of CNR2 knockout mice suggests that the loss of CNR2 may have a profound impact on T cell homeostasis and function in EM dependent manner, as this was not the case in sham-operated mice. This impairment in T cell function has direct consequences on the adaptive immune response, as evidenced by the altered gene expression profiles and pathways related to immune function in our transcriptomic analysis. But we haven't conducted specific mechanistic experiments to tease out the interplay between immune response and endometriosis lesion development in our model systems.

Several limitations should be acknowledged in our study. Firstly, understanding the homeostatic aspects of ECS, both with and without the presence of CNR1 and CNR2, remains a complex challenge. While our global k/o mouse models provide valuable insights, further research utilizing targeted k/o specific to the uterus could offer a more precise understanding of their contributions to uterine physiology and further implications in EM establishment. The use of mouse models to study EM has inherent limitations due to species differences and the inability to fully recapitulate the human disease. The molecular and cellular complexities associated with DD and UnD endometrial tissues, as well as the timing of EM induction in these mice, may not perfectly mirror the human condition. No significant baseline alterations were observed in the immune profile between the CNR1 and CNR2 k/o and the WT mice without EM induction. However, the uterine environment has not been assessed to understand the baseline immune profile between the k/o and WT mice. These limitations emphasize the need for future investigations to enhance the translational relevance of our findings and further our understanding of the complex interplay between ECS, decidualization, and EM pathogenesis.

In conclusion, our study offers evidence for the involvement of CNR1 and CNR2 dysregulation in EM pathogenesis. Through an integrative analysis of transcriptomic profiles, immune cell dynamics, and spatial relationships within EM lesions from mice, we unveil the intricate interactions between ECS, immune responses, and cellular changes in EM. By identifying potential mechanisms through which ECS disruption could impact EM, our research provides a foundation for the development of targeted therapies addressing the ECS's influence on EM. These findings will advance our understanding of EM and lead to innovative therapeutic strategies to manage this complex disorder.

## Methods
### Animals
Experiments described in this work were approved by the Queen's University Institutional Animal Care Committee as per the guidelines provided by the Canadian Council of Animal Care for protocol number 2021–2228. All animals were assigned randomly to the surgical procedures. All studies were performed using adult female mice at ages between 7–10 weeks. CNR1 k/o (B6.129P2(C)-*Cnr1*tm1.1Ltz/J) and CNR2 k/o (B6.129P2-*Cnr2*tm1Dgen/J) male and female breeder mice were obtained from Jackson Laboratory (Bar Harbor, USA) and were housed in the Queen's University animal facility. CNR1 k/o and CNR2 k/o experimental female mice were obtained by trio breeding with their respective homozygous k/o male counterparts. C57BL/6 j (WT) control female mice and vasectomized male mice at 7–10 weeks were obtained from Jackson Laboratory. All breeder mice were housed in standard breeding cages in a barrier facility and the experimental animals were housed in a conventional holding area. Animals were housed at constant temperature (22 ±1°C) and relative humidity (50%),

with a 12:12 hr light: dark cycle (light on 07.00–19.00 hr). Food and water were available ad libitum. All experimental animals were acclimatized at the conventional housing facility for 1 week before starting the experiments.

## In vivo decidualization

In this study, we have used a modified syngeneic mouse model of EM, where the donor fragments were obtained from artificially decidualized uterine horns. The method of artificial decidualization used in this study has been previously established and utilized by several research studies (*Cai et al., 2022*; *Benson et al., 1996*; *Liu et al., 2022*). To artificially induce decidualization, female mice were allowed to mate with vasectomized male mice to induce pseudopregnancy. After day 4 of pseudo-pregnancy, female mice were subjected to laparotomy to receive a 30 µL injection of sesame seed oil (S3547, Sigma, USA), intra-luminally into one uterine horn to induce DD. The contralateral, uninjected horn served as an UnD control. After the sesame seed oil injection, animals were rested for 4 days, after which the DD was successfully induced in one uterine horn as shown in *Figure 1A*. These uterine horns were utilized as donor fragments to induce EM in recipient mice of their respective genotype. *Figure 1B* shows the representative images of EM lesions 7 days post-surgical induction.

## Mouse model of EM

EM was surgically induced as described previously (*Lingegowda et al., 2021b*; *Lingegowda et al., 2021a*). Two independent groups (DD and UnD) per genotype were used in this study (n=8–16). Briefly, the DD and UnD uterine horns from the donor mice were harvested, and uterine horns were longitudinally dissected to reveal the endometrium. Uterine fragments were obtained using a 3.0 mm epidermal biopsy punch (33–32, Integra Miltex, USA). Recipient mice were anesthetized under 3.5% isoflurane vaporizer anesthesia to make a midline incision in the abdomen (n=8–16) and two 3.0 mm DD or UnD uterine fragments were implanted on the right inner peritoneal wall using a veterinary grade tissue bonding glue (1469 SB, 3 M, USA). WT, CNR1 k/o, and CNR2 k/o control groups (n=4) were sham-operated with a midline incision in the abdomen without implantation of uterine frag-ments. Mice were sacrificed 7 days after EM induction surgery since the focus of the study was on the earlier time point of EM initiation after induction. Based on the previous studies, 7 days after EM induction appears to be the log phase of tissue repair and immune response (*Symons et al., 2020*). Blood was harvested through cardiac puncture to assess EC ligands. Peritoneal fluid (PF) was collected by injecting 3 ml of ice-cold phosphate-buffered saline (PBS) into the peritoneal cavity. Spleens were collected in ice-cold RPMI media (11875093, ThermoFisher, Canada) before processing to obtain single-cell suspension. EM lesions were either snap-frozen in liquid nitrogen and stored at –80°C or processed using 4% paraformaldehyde overnight (12–20 hr), kept at 4 °C in 70% ethanol, and then embedded in paraffin.

## Lipid extraction and targeted mass spectrometry

Plasma was undiluted and ~10 mg of tissue per sample were homogenized with RIPA buffer (89900, Thermo Fisher, Canada) with 1:100 of protease inhibitor cocktail (535140–1 ML, Sigma, Canada) to obtain tissue lysates. Both plasma and tissue lysates obtained were individually subjected to solid phase extraction (SPE). Internal standards, both deuterated and non-deuterated used in the study to assess the ECS ligands were purchased from Cayman Chemicals, USA (*Supplementary file 1a*). Aliquots of 100 µL of plasma and tissue lysates were added to a protein precipitation plate (CE0-7565-R, Phenomenex, USA) along with 200 µL of cold acetonitrile containing the deuterated internal standards. The filtrate was diluted with 500 µL of water and submitted to SPE extraction on an Oasis HLB 96-Well Plate (WAT058951, Waters, Canada). Samples were washed with 60% methanol prior to elution with acetonitrile. The eluate was dried, reconstituted in 100 µL of mobile phase A and analyzed by liquid chromatography-mass spectrometry (LC-MS as described in *Supplementary file 1b*). The endogenous concentration for the four compounds in human plasma were calculated by standard addition.

## Flow cytometry

Single-cell suspensions were prepared from murine spleens by mechanical dissociation, RBC lysis, and centrifugation. Splenic cells and cells from PF were resuspended in a staining buffer (PBS with

2% fetal bovine serum) at a concentration of 0.5×10^6 cells/mL. All antibodies used for flow cytometry analyses were purchased from BioLegend, USA, unless otherwise mentioned. The antibodies included CD45-FITC (103107), CD3-BV510 (100234), CD4-BV785 (100551), CD8-BV605 (100744), CD11b-AF700 (101222), F4/80-PE/Cy7 (123114), NK1.1-APC/Cy7 (108724), and CD19-PE/Dazzle 594 (115554). Staining was performed following the manufacturer's recommendations. For each sample, 50 µL of the antibody cocktail was added to 50 µL of cell suspension in 96-well plates. The mixture was incubated at 4 °C for 20 min in the dark, along with an anti-CD16/32 Fc block antibody (101319). After incubation, cells were washed twice with staining buffer and centrifuged before fixing the cells with fixation buffer (00-8222/49, Thermo Fisher, Canada). Flow cytometry analysis was carried out using a Beckman Coulter CytoFlex S flow cytometer. Compensation controls were established using single antibody-stained cells. Isotype controls provided baseline levels of non-specific staining and cell populations were defined using fluorescence minus one (FMO) control. Data analysis employed FlowJo software (v 10.9; FlowJo, USA) as well as SPECTRE (v 1.0) computational toolkit in R (v 4.2.3) to obtain t-distributed stochastic neighbor embedding (t-SNE) plots based on the unsupervised flowSOM clusters generated by marker expression.

### In-vitro T cell functional assay

Total CD3 + T cells were isolated from splenocytes of naive WT and CNR2 k/o mice using a negative selection magnetic kit (19851 A, StemCell, Canada) following the manufacturer's instructions. All recombinant proteins and compounds were purchased from Biolegend, USA, unless otherwise mentioned. Subsequently, 250,000T cells per well were seeded into a 96-well plate coated with anti-mouse CD3 [(2 µg/ml), (100340)]. RPMI-1640 media supplemented with rmIL-2 [(10 ng/ml), (575404)], anti-mouse CD28 [(5 µg/ml), (102116)], 10% fetal bovine serum, β-mercaptoethanol (50 µM), and penicillin/streptomycin (100 U/ml) was used as the growth medium. T cells were then activated non-specifically with or without the cell activation cocktail consisting of Phorbol 12-myristate 13-acetate (PMA) and ionomycin [(50 ng/ml PMA and 1 µM ionomycin), (423301)] in the presence and absence of TNFα [(100 ng/ml), (410-MT-010/C, R&D Systems, USA)] to simulate a sterile inflammatory challenge. Following a 48 hr incubation period, brefeldin A [(10 µg/ml), (11861, Cayman Chemicals, USA)] was introduced to the cells to measure intracellular interferon-gamma (IFNγ) levels at the 42 hr time point. Flow cytometry analysis was conducted using a panel of markers, purchased from Biolegend, USA unless otherwise mentioned, including CD3e-FITC (100306), CD4-AF700 (100430), CD8-PE/fire700 (100792), Ki67-PB (151223), FoxP3-PE (126404), IFNγ -BV605 (505840), and Live/dead-K0525 (L304966, Thermo Fisher, Canada), to assess various T cell subsets and viability. Flow cytometry staining and acquisition was carried out as described above with the addition of permeabilization (00-8333-56, Thermo Fisher, Canada) buffer to stain for intracellular markers (Ki-67 and IFNγ), according to the manufacturer's instructions. Data analysis was performed using FlowJo software and visualized using GraphPad Prism (v 9.5.1).

### RNA isolation using RNeasy mini kit

Snap-frozen UnD and DD EM lesions from WT, CNR1 k/o, and CNR2 k/o were homogenized, and RNA was isolated using the RNeasy Mini Kit (74104, Qiagen, Canada), according to the manufacturer's instructions. Briefly, ~20 mg EM lesion tissues were placed individually in PowerBead Ceramic Tubes (13113–50, Qiagen, Canada) along with lysis buffer. Lesions were homogenized using a Bead Ruptor homogenizer (Omni International, USA) and lysates were extracted after centrifugation at 10,000 RCF. Lysate was mixed with 70% ethanol, added to the RNeasy spin column, and then centrifuged to bind the RNA to the column. Spin column was washed twice, and RNA was isolated using an elution buffer. Total RNA quality was measured using the nanodrop spectrophotometer and stored at –80°C before shipping to BGI Global (Boston, USA) for bulk RNA analysis. RNA integrity was determined using the Agilent 4150 TapeStation System Agilent, USA for sample quality control, and only samples with RNA quality number ≥7 were considered for library preparation and further sequencing.

### RNA library preparation, sequencing, and analysis

Library preparation began with mRNA enrichment using oligo dT beads, which selectively capture mRNA molecules. Next, the enriched mRNA was fragmented, and first-strand cDNA was synthesized using random N6 primers, followed by second-strand cDNA synthesis using deoxyuridine triphosphate

(dUTP). After cDNA synthesis, end repair was performed to generate blunt ends, and 3' adenylation was carried out to facilitate adaptor ligation. Adaptors were ligated to the 3' adenylated cDNA fragments. To enrich the cDNA library for sequencing, PCR amplification was conducted. Prior to amplification, the dUTP-marked strand was specifically degraded by Uracil-DNA-Glycosylase (UDG). The remaining first-strand cDNA was then amplified using PCR primers. Following amplification, single-strand separation was achieved through denaturation by heat. The single-stranded DNA was cyclized using a splint oligo and DNA ligase. DNA nanoball synthesis was performed on the cyclized single-stranded DNA templates. This process facilitated the generation of clonal DNA clusters, providing the material necessary for subsequent sequencing. Sequencing was executed using the DNBSEQ Technology platform. The prepared DNA libraries were loaded onto the DNBSEQ sequencer and sequenced at an average depth of 30 million paired-end reads (2 × 100) per library. The sequencing data was filtered with SOAPnuke by removing reads containing sequencing adapter; removing reads whose low-quality base ratio (base quality less than or equal to 15) is more than 20%, and removing reads whose unknown base ('N' base) ratio is more than 5%. Next, clean reads were obtained and stored in FASTQ format. Clean reads were mapped to the mouse reference genome (NCBI: GRCm38. p6) using HISAT2 (v2.0.4). The subsequent analysis and data mining were performed on Dr. Tom Multiomics Data mining system (https://biosys.bgi.com).

## Imaging mass cytometry: Labeling

A comprehensive panel of antibodies identifying innate and adaptive immune cell populations and cell types that are integral to the EM lesion microenvironment (*Supplementary file 1c*) was designed and optimized as previously described (*McDowell et al., 2021*; *Sorin et al., 2023*). The formalin-fixed paraffin-embedded (FFPE) tissue sections (n=2–3 per tissue type) underwent deparaffinization and heat-mediated antigen retrieval on the Ventana Discovery Ultra auto-stainer platform (Roche Diagnostics, Canada), following the below instructions. Initially, the slides were exposed to a temperature of 70°C in a pre-formulated EZ Prep solution (Roche Diagnostics, Canada), followed by a subsequent incubation at 95°C in pre-formulated Cell Conditioning 1 solution (Roche Diagnostics, Canada). Following this, the slides were washed in 1x PBS and then exposed to Dako Serum-free Protein Block solution (Agilent, USA) for 45 min at room temperature. An antibody cocktail, containing metal-conjugated antibodies, was prepared using Dako Antibody Diluent (Agilent, USA) at specified dilutions. The primary antibodies within this cocktail were applied to the slides and left to react overnight at 4°C, after which the slides were washed with 0.2% Triton X-100 and 1x PBS. For the subsequent step, a secondary antibody cocktail comprising metal-conjugated anti-biotin antibodies was created in Dako Antibody Diluent, at a predetermined dilution. The slides were treated with this anti-biotin cocktail for 1 hr at room temperature and then washed with 0.2% Triton X-100 and 1x PBS. For counterstaining, the slides were exposed to Cell-ID Intercalator-Ir (Fluidigm, Canada) diluted at a ratio of 1:400 in 1x PBS for 30 min at room temperature. After a 5 min rinse with distilled water, the slides were air-dried in preparation for IMC acquisition. The Hyperion Imaging System (Fluidigm, Canada) was employed for the IMC acquisition process.

## Imaging mass cytometry: Data analysis

Lesions (n=2–3 per tissue type) were grouped based on the origin of uterine fragments (i.e. UnD or DD) from three different genotypes (WT, CNR1 k/o, and CNR2 k/o). IMC data analysis methods employed in this study follow established procedures as outlined in the Steinbock toolkit (Spatial Experiment v 1. 12. 0) for data preprocessing, image segmentation, and object quantification (*Windhager et al., 2021*). Cell segmentation utilized a deep learning approach described by *Greenwald et al., 2022*. Briefly, dual-channel images were generated using nuclear and cytoplasmic markers, representing respective signals. The DeepCell tool with Mesmer, a pre-trained deep learning segmentation algorithm from TissueNet, was used to automate cell mask generation, requiring no additional user input. Given the IMC data was acquired in batches, we performed batch effect corrections using the harmony algorithm as described (*Korsunsky et al., 2019*). This involved iterative clustering and correction of cell positions in the principal component analysis (PCA) space. Subsequently, unsupervised PhenoGraph clustering in R (v 4. 3. 2) was used to categorize cell types. For this, signals including αSMA, B220, CD19, β-catenin, CD3, CD4, CD8, CD11b, CD11c, CD31, CD68, E-cadherin, MPO, pan-cytokeratin, and vimentin were utilized, employing a k-value of 60. To ascertain cell interactions,

imcRtools (v 1. 8. 0) and cytomapper (v 1. 14. 0) in R were employed for visualization. A permutation test evaluated interactions with neighboring cells. Neighboring cells were defined as those within a five-pixel radius (5 μm), and the buildSpatialGraph function established the number of one cluster neighbors interacting with another cluster. A default of 1000 permutations was set. Each iteration led to an interaction score and p-value computation, and the significant outcomes (at alpha 1% risk) were depicted in heatmaps. To delineate spatial cellular neighborhoods, neighbor windows were computed, representing the N nearest cells to each cell. This process followed previous protocols. Employing imcRtools, cellular neighborhood grouping was conducted, leading to the identification of 8 cellular neighborhoods in the lesions.

## Statistics

Statistical analyses performed to compare the concentration of EC ligands through targeted LC-MS and evaluation of the immune cell population via flow cytometry were conducted using Prism GraphPad. A one-way analysis of variance (ANOVA) was performed with the Holm-Sidak post-hoc test to determine the specific pairwise differences between the groups. For in-vitro T cell functional assay, two-way ANOVA was performed using Tukey's post-hoc test to compare within and between the groups. The significance level was set at $\alpha$=0.05. Data are presented as mean ± standard deviation (SD) unless otherwise stated.

## Acknowledgements

We thank Dr. Alexandra Furtos and Karine Gilbert from the Regional Mass Spectrometry Centre at Université de Montréal for designing and performing mass-spectrometry evaluation; Dr. Yuhong Wei and the single cell and imaging mass cytometry platform (SCIMAP) at McGill University Goodman Cancer Institute for processing, labeling, and acquiring samples for IMC imaging; Brittney Armitage-Brown from the Animal Care Services at Queen's University for breeding mice utilized in this study. This work was supported by funding from the Canadian Institutes of Health Research (CIHR-394340) to CT, and MK. This research was supported with funds from Canadian Institutes of Health Research (CIHR).

## Additional information

### Funding

| Funder | Grant reference number | Author |
|---|---|---|
| Canadian Institutes of Health Research | 394340 | Chandrakant Tayade |

The funders had no role in study design, data collection and interpretation, or the decision to submit the work for publication.

### Author contributions

Harshavardhan Lingegowda, Conceptualization, Data curation, Formal analysis, Validation, Investigation, Visualization, Methodology, Writing - original draft, Writing - review and editing; Katherine B Zutautas, Yuhong Wei, Data curation, Methodology; Priyanka Yolmo, Data curation, Investigation, Methodology; Danielle J Sisnett, Alison McCallion, Methodology; Madhuri Koti, Resources, Funding acquisition; Chandrakant Tayade, Conceptualization, Resources, Funding acquisition, Writing - review and editing

### Author ORCIDs

Harshavardhan Lingegowda  http://orcid.org/0000-0001-6030-4936
Katherine B Zutautas  https://orcid.org/0000-0003-4304-1024
Danielle J Sisnett  https://orcid.org/0000-0001-8061-0631
Chandrakant Tayade  https://orcid.org/0000-0001-9062-050X

### Ethics

Experiments described in this work were approved by the Queen's University Institutional Animal Care Committee (protocol # 2021-2228) as per the guidelines provided by the Canadian Council of Animal Care.

Reviewer #1 (Public Review): https://doi.org/10.7554/eLife.96523.3.sa1
Reviewer #2 (Public Review): https://doi.org/10.7554/eLife.96523.3.sa2
Author response https://doi.org/10.7554/eLife.96523.3.sa3

## Additional files

### Supplementary files

• Supplementary file 1. Deuterated and non-deuterated standards, experimental LC-MS conditions for ECS ligand identification, and IMC antibodies with dilutions.

• Supplementary file 2. List of total differentially expressed genes in the lesions of wild-type (WT), CNR1 k/o, and CNR2 k/o mice.

• Supplementary file 3. Differentially expressed immune related genes in the lesions of wild-type (WT), CNR1 k/o, and CNR2 k/o mice.

• MDAR checklist

• Source data 1. Normalized, total bulk RNA sequencing data along with sample ID.

### Data availability

Bulk mRNA sequencing data is provided in the supplementary files (Supplementary Data 4) and IMC data generated in this study has been deposited in Mendeley Data (https://doi.org/10.17632/2ptns5yhzh.2). All the codes used in the current study are from previously published articles, as cited in the text. Authors do not report original code.

The following dataset was generated:

| Author(s) | Year | Dataset title | Dataset URL | Database and Identifier |
| --- | --- | --- | --- | --- |
| Lingegowda H, Tayade C | 2024 | Mouse Endometriosis lesion IMC Dataset Lingegowda et al 2024 | https://doi.org/10.17632/2ptns5yhzh.2 | Mendeley Data, 10.17632/2ptns5yhzh.2 |

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
