## [Editor Report · eLife assessment]

This study presents **valuable** findings on how the endocannabinoid system is involved in endometriosis progression using CNR1 and CNR2 knockout (KO) mouse models. The evidence supporting the authors' claims is **incomplete**; including bulk RNA-seq, flow cytometry, and imaging mass cytometry would have strengthened the study. This work might be of interest to medical scientists working on endometriosis.

---

## [Referee Report · Reviewer #1 (Public Review)]

Summary:

The endocannabinoid system (ECS) components are dysregulated within the lesion microenvironment and systemic circulation of endometriosis patients. Using endometriosis mouse models and genetic loss of function approaches, Lingegowda et al. report that canonical ECS receptors, CNR1 and CNR2, are required for disease initiation, progression, and T-cell dysfunction.

Strengths:

The approach uses genetic approaches to establish in vivo causal relationships between dysregulated ECS and endometriosis pathogenesis. The experimental design incorporates both bulk and single-cell RNAseq approaches, as well as imaging mass spectrometry to characterize the mouse lesions. The identification of immune-related and T-cell-specific changes in the lesion microenvironment of CNR1 and CNR2 knockout (KO) mice represents a significant advance

Weaknesses:

Although the mouse phenotypic analyses involves a detailed molecular characterization of the lesion microenvironment using genomic approaches, detailed measurements of lesion size/burden and histopathology would provide a better understanding of how CNR1 or CNR2 loss contributes to endometriosis initiation and progression. The cell or tissue-specific effects of the CNR1 and CNR2 are not incorporated into the experimental design of the studies. Although this aspect of the approach is recognized as a major limitation, global CNR1 and CNR2 KO may affect normal female reproductive tract function, ovarian steroid hormone levels, decidualization response, or lead to preexisting alterations in host or donor tissues, which could affect lesion establishment and development in the surgically induced, syngeneic mouse model of endometriosis.

---

## [Referee Report · Reviewer #2 (Public Review)]

Summary:

The endocannabinoid system (ECS) regulates many critical functions, including reproductive function. Recent evidence indicates that dysregulated ECS contributes to endometriosis pathophysiology and microenvironment. Therefore, the authors further examined the dysregulated ECS and its mechanisms in endometriosis lesion establishment and progression using two different endometrial sources of mouse models of endometriosis with CNR1 and CNR2 knockout mice. The authors presented differential gene expressions and altered pathways, especially those related to the adaptive immune response in CNR1 and CNR2 ko lesions. Interstingly, the T-cell population was dramatically reduced in the peritoneal cavity lacking CNR2, and the loss of proliferative activity of CD4+ T helper cells. Imaging mass cytometry analysis provided spatial profiling of cell populations and potential relationships among immune cells and other cell types. This study provided fundamental knowledge of the endocannabinoid system in endometriosis pathophysiology.

Strengths:

Dysregulated ECS and its mechanisms in endometriosis pathogenesis were assessed using two different endometrial sources of mouse models of endometriosis with CNR1 and CNR2 knockout mice. Not only endometriotic lesions but also peritoneal exudate (and splenic) cells were analyzed to understand the specific local disease environment under the dysregulated ECS.

Providing the results of transcriptional profiles and pathways, immune cell profiles, and spatial profiles of cell populations support altered immune cell population and their disrupted functions in endometriosis pathogenesis via dysregulation of ECS.

L386: Role of CNR2 in T cells: Finding nearly absent CD3+ T cells in the peritoneal cavity of CNR2 ko mice is intriguing.

Interpretation of the results is well-described in discussion.

Weaknesses:

The study was terminated and characterized 7 days after EM induction surgery without the details for selecting the time point to perform the experiments.

The authors also mentioned that altered eutopic endometrium contributes to the establishment and progression of endometriosis. This reviewer agrees L324-325. If so, DEGs are likely identified between eutopic endometrium (with/without endometriosis lesion induction) and ectopic lesions. It would be nice to see the data (even though using publicly available data sets).

Figure 7 CDEF. Please add the results of the statistical analyses and analyzed sample numbers. L444-450 cannot be reviewed without them.

This reviewer agrees L498-500. In contrast, retrograded menstrual debris is not decidualized. The section could be modified to avoid misunderstanding.

The authors addressed all my concerns. I do not have any comments.

---

## [Author Response]

The following is the authors’ response to the original reviews.

**Public Reviews:**

**Reviewer #1 (Public Review):**
Summary:The endocannabinoid system (ECS) components are dysregulated within the lesion microenvironment and systemic circulation of endometriosis patients. Using endometriosis mouse models and genetic loss of function approaches, Lingegowda et al. report that canonical ECS receptors, CNR1 and CNR2, are required for disease initiation, progression, and T-cell dysfunction.Strengths:The approach uses genetic approaches to establish in vivo causal relationships between dysregulated ECS and endometriosis pathogenesis. The experimental design incorporates both bulk and single-cell RNAseq approaches, as well as imaging mass spectrometry to characterize the mouse lesions. The identification of immune-related and T-cell-specific changes in the lesion microenvironment of CNR1 and CNR2 knockout (KO) mice represents a significant advanceWeaknesses:Although the mouse phenotypic analyses involve a detailed molecular characterization of the lesion microenvironment using genomic approaches, detailed measurements of lesion size/burden and histopathology would provide a better understanding of how CNR1 or CNR2 loss contributes to endometriosis initiation and progression. The cell or tissue-specific effects of the CNR1 and CNR2 are not incorporated into the experimental design of the studies. Although this aspect of the approach is recognized as a major limitation, global CNR1 and CNR2 KO may affect normal female reproductive tract function, ovarian steroid hormone levels, decidualization response, or lead to preexisting alterations in host or donor tissues, which could affect lesion establishment and development in the surgically induced, syngeneic mouse model of endometriosis.

We appreciate the reviewer's thoughtful and constructive feedback. We agree that the additional measurements of lesion size/burden and histopathology would provide valuable insights into the specific contributions of CNR1 and CNR2 to endometriosis progression. However, the focus of this study was on assessing the alterations in complex immune microenvironment due to the absence of CNR1 and CNR2, given their close relation in regulating immune cell populations. We will plan to incorporate these measurements in future studies to further strengthen the understanding of the disease pathogenesis. Regarding the potential effects of global knockout, the reviewer raises a valid concern. To address this, we will explore cell and/or tissue-specific knockout models in future experiments to better isolate the direct effects of CNR1 and CNR2 on the disease process, while minimizing potential confounding factors from systemic alterations.

**Reviewer #2 (Public Review):**
Summary:The endocannabinoid system (ECS) regulates many critical functions, including reproductive function. Recent evidence indicates that dysregulated ECS contributes to endometriosis pathophysiology and the microenvironment. Therefore, the authors further examined the dysregulated ECS and its mechanisms in endometriosis lesion establishment and progression using two different endometrial sources of mouse models of endometriosis with CNR1 and CNR2 knockout mice. The authors presented differential gene expressions and altered pathways, especially those related to the adaptive immune response in CNR1 and CNR2 ko lesions. Interestingly, the T-cell population was dramatically reduced in the peritoneal cavity lacking CNR2, and the loss of proliferative activity of CD4+ T helper cells. Imaging mass cytometry analysis provided spatial profiling of cell populations and potential relationships among immune cells and other cell types. This study provided fundamental knowledge of the endocannabinoid system in endometriosis pathophysiology.Strengths:Dysregulated ECS and its mechanisms in endometriosis pathogenesis were assessed using two different endometrial sources of mouse models of endometriosis with CNR1 and CNR2 knockout mice. Not only endometriotic lesions, but also peritoneal exudate (and splenic) cells were analyzed to understand the specific local disease environment under the dysregulated ECS.Providing the results of transcriptional profiles and pathways, immune cell profiles, and spatial profiles of cell populations support altered immune cell population and their disrupted functions in endometriosis pathogenesis via dysregulation of ECS.In line 386: Role of CNR2 in T cells. The finding that nearly absent CD3+ T cells in the peritoneal cavity of CNR2 ko mice is intriguing.The interpretation of the results is well-described in the Discussion.Weaknesses:The study was terminated and characterized 7 days after EM induction surgery without the details for selecting the time point to perform the experiments.The authors also mentioned that altered eutopic endometrium contributes to the establishment and progression of endometriosis. This reviewer agrees with lines 324-325. If so, DEGs are likely identified between eutopic endometrium (with/without endometriosis lesion induction) and ectopic lesions. It would be nice to see the data (even though using publicly available data sets).Figure 7 CDEF. The results of the statistical analyses and analyzed sample numbers should be added. Lines 444-450 cannot be reviewed without them.This reviewer agrees with lines 498-500. In contrast, retrograded menstrual debris is not decidualized. The section could be modified to avoid misunderstanding.

We would like to thank the reviewer for insightful comments, suggestions and acknowledging the importance of the work presented in this manuscript.

Regarding 7-day time point, we have provided rationale in lines 479-481, but agree that it isn’t sufficient and hence we have provided additional details on the selection of the 7-day time point for the experiments in methods section (Mouse model of EM). We have also noted the suggestion on providing comparison of differentially expressed genes in the eutopic endometrium vs ectopic lesions. Since there are publications comparing the eutopic vs ectopic gene expression patterns (PMIDs: 33868805 and 18818281), including a study exploring the ECS genes in the endometrium throughout different menstrual cycles (PMID: 35672435), we believe additional analysis using the same dataset may not yield new information. However, we see the value in reviewer’s comment, and we will look at the gene expression patterns in the uterine vs endometriosis like lesions in our future studies with tissue or cell specific CNR1 and CNR2 knockout models to understand functional relevance of ECS in endometriosis initiation.

Since the IMC study was exploratory for proof of concept, we did not have enough biological replicates for meaningful statistical validation (n = 2-3). We have clarified this information in the methods, results, and figure legends for appropriately representing the limitations of the current setup.

Finally, we appreciate the feedback on the section discussing retrograded menstrual debris. Even though the menstrual debris may not be decidualized, some endometriotic lesions have the ability to decidualize based on their response to estrogen and progesterone in a cycling manner (PMID: 26450609), similar to the endometrium in the uterine cavity. We have clarified this in the revised MS.

**Recommendations for the Authors:**

**Reviewer #1 (Recommendations For The Authors):**
The mechanism of how alterations in ECS contribute to the observed cellular and molecular changes is unclear. Connecting CNR1 or CNR2 function to a specific cell type or cellular process would provide a more detailed understanding of how dysregulated ECS contributes to endometriosis pathogenesis.

We agree that integrating the functions of CNR1 or CNR2 to specific cell types or cellular processes would strengthen the mechanistic insights presented in our study. This would help elucidate specific pathways by which dysregulated ECS leads to the alterations in immune cell populations, gene expression profiles, and other key aspects of endometriosis development and progression. This is a rapidly evolving field and at this stage, we do not have published information to reflect on this aspect in the revised manuscript.

(1) As mentioned in the text, the ECS components being studied are widely expressed and may affect multiple aspects of endometriosis pathogenesis and symptomatology. However, the cell or tissue-specific effects of the CNR1 and CNR2 are not incorporated into the experimental design of the studies. Although these limitations are mentioned in the discussion, it is important to know if global CNR1 and CNR2 KO affect normal female reproductive tract function, ovarian steroid hormone levels, decidualization response, or if preexisting alterations in host or donor tissues affect lesion development in the surgically induced, syngeneic mouse model of endometriosis. This would also be the case in studies on immune system dysfunction or lesion microenvironment, as it is possible preexisting immune system dysfunction following CNR1 or CNR2 loss could alter the disease trajectory and lead to a misinterpretation of the findings. Some of these potential confounders could be addressed using crossover approaches in Figure 1A experimental design, but the donor tissues are reported to be matched to the recipients based on genotype.

The reviewer raised an excellent point that the widespread expression of the ECS components studied in our manuscript may affect multiple aspects of endometriosis pathogenesis and symptomatology. Indeed, the cell or tissue-specific effects of CNR1 and CNR2 knockout are not fully incorporated into our experimental design, which could lead to potential confounding factors that may affect the interpretation of some of our findings. However, as outlined in our previous comments, we will incorporate the tissue/cell specific knockout, as well the crossover approaches to elucidate if the loss of CNR1 and CNR2 function is lesion driven in future studies. We agree that it is important to understand the impact of global CNR1 and CNR2 knockout on normal female reproductive tract function, ovarian steroid hormone levels, decidualization response, and other potential preexisting alterations in the host or donor tissues that could influence lesion development in the syngeneic mouse model of endometriosis. As outlined in the MS (lines 59-62), there are studies highlighting pregnancy specific impact including implantation and impaired primary decidual zone formation. We did not find any baseline alterations in the systemic immune profiles between the CNR1 and CNR2 knockout mice and the WT mice without EM induction. However, the uterine environment has not been assessed to understand the baseline immune profile between the knockout mice and WT mice. We agree with the reviewer that, the possibility of preexisting immune system dysfunction following CNR1 or CNR2 loss could alter the disease trajectory related to immune system dysfunction or lesion microenvironment. We have highlighted this in the limitations section.

(2) The phenotypic characterization of the endometriosis mouse model with or without CNR1 or CNR2 KO is very limited. To better understand how the observed cellular and molecular alterations correlate with endometriosis pathogenesis and severity CNR1 and CNR2 K/O mice, a detailed characterization of lesion size differences and histopathology should be made. Importantly, the histopathological characterization of the lesions would complement the imaging mass spectrometry findings.

We agree that more detailed characterization of the endometriosis lesions in our CNR1 and CNR2 knockout mouse models are required. As evident for our several previous publications, we have focused on detailed histopathological characterization of endometriotic lesions in our syngeneic mouse model of endometriosis including a multiple time course study (Symons et al, 2020, FASEB). In the present investigation, we focused on cataloging spatial and transcriptomic changes as we do not currently have any information on the global influence of CNR1 and CNR2 knockout on endometriosis lesion microenvironment, since we prioritized this aspect, we were not able to provide detailed histological assessment of lesions. However, the IMC analysis provides a detailed, spatially resolved profile of the cellular composition and interactions within the endometriotic lesions, which we believe offers valuable insights into the mechanisms by which the dysregulated ECS may contribute to endometriosis pathogenesis. This quantitative, high-dimensional approach complements the transcriptional profiling and other analyses we have performed.

(3) Given the effect sizes and variance observed with the ECS ligand measurements, an N = 4-5 biological samples for mouse phenotypic studies seems too low.

The reviewer raises a valid point about low sample size. As elaborated earlier, this was a proof of principle study to capture biologically significant alterations within lesion and surrounding peritoneal microenvironment in the absence of CNR1, CNR2 receptors. This information is crucial for establishing the potential mechanisms by which the dysregulated ECS may contribute to the pathogenesis of endometriosis. Now that we have established the framework and baseline understanding of immune-inflammatory alterations, we will refine our future experimental approaches and include more samples if becomes necessary.

**Reviewer #2 (Recommendations For The Authors):**
It is hard to read the labeling of figures. Please increase the font size of each figure.

We have increased the font size of the labels where necessary to improve the readability.

Supplementary Data 1, Table 1 seems like Supplementary Table 1. Please use the same labeling of the Supplementary tables and figures to avoid confusion.

We have updated the labeling accordingly and ensured that all supplementary tables and figures are consistently labeled.

This reviewer suggests depositing RNA-seq and IMC data to NCBI etc. and listing the accession number in the MS.

Thank you for your recommendation to deposit the RNA-seq and imaging mass cytometry (IMC) data from our study in public repositories such as NCBI. We appreciate your suggestion, as data sharing is an important aspect of scientific transparency and reproducibility. Bulk mRNA sequencing data has been attached as a supplementary file and IMC data has been deposited on Mendeley Data (DOI: 10.17632/2ptns5yhzh.1).

Please clarify L363.

We have clarified this in the revised MS. The revised text now reads: “However, we did not find the same differences (T cell-related genes) in the UnD lesions of CNR2 k/o mice. Moreover, UnD lesions of CNR2 k/o mice showed significantly low number of DEGs (11 compared to 65 in the DD lesions from CNR2 k/o mice) suggesting a decidualization dependent response (Supplementary Data 3).”

Figure 7B: It is hard to see/understand the results in L438-440. It might be helpful if % is added to the figure.

We have added more tick marks to the y-axis of Figure 7B to make it easier for the reader to interpret the percentages of the different cell types.

Figure 7 legend: 2nd D should be G.

We have revised the legend accordingly.

Supplementary Figure 6: It seems immune cells are clustered in CN1, which is different from Figure 7. To easily understand Suppl Fig 6AB, please add some details in the legend.

We have revised the legend as suggested.

The revised legend now reads: “A, B Representative image of 8 distinct cell types from CN analysis of DD and UnD lesions from WT, CNR1 k/o, and CNR2 k/o mice, respectively. C Heatmap representation of CN analysis shows distinct clustering patterns observed in the UnD lesions among the different genotypes. The clustering reveals distinct spatial patterns of immune cell populations within the UnD lesions, which appear to differ from the observations in Figure 7G. This suggests potential spatial heterogeneity in the immune landscape of EM like lesions under conditions of decidualization.”